# De novo apical domain formation inside the *Drosophila* adult midgut epithelium

Jia Chen, Daniel St Johnston*

The Gurdon Institute, University of Cambridge, Cambridge, United Kingdom

**Abstract** In the adult *Drosophila* midgut, basal intestinal stem cells give rise to enteroblasts that integrate into the epithelium as they differentiate into enterocytes. Integrating enteroblasts must generate a new apical domain and break through the septate junctions between neighbouring enterocytes, while maintaining barrier function. We observe that enteroblasts form an apical membrane initiation site (AMIS) when they reach the septate junction between the enterocytes. Cadherin clears from the apical surface and an apical space appears between above the enteroblast. New septate junctions then form laterally with the enterocytes and the AMIS develops into an apical domain below the enterocyte septate junction. The enteroblast therefore forms a pre-assembled apical compartment before it has a free apical surface in contact with the gut lumen. Finally, the enterocyte septate junction disassembles and the enteroblast/pre-enterocyte reaches the gut lumen with a fully formed brush border. The process of enteroblast integration resembles lumen formation in mammalian epithelial cysts, highlighting the similarities between the fly midgut and mammalian epithelia.

## Editor's evaluation

This paper addresses a fundamental cell biological question and will be of interest to a wide range of researchers including those in the fields of cell, development, stem cell and cancer biology. The finding that newborn cells (enterocytes) in the *Drosophila* midgut form a pre-apical compartment containing a fully-formed brush border prior to integrating into the epithelium, which is described in great detail, is highly novel and exciting. This work highlights similarities between fly midgut polarization and lumen formation in mammalian epithelial cysts.

*For correspondence:
d.stjohnston@gurdon.cam.ac.uk

**Competing interest:** The authors declare that no competing interests exist.

## Introduction

Like the mammalian gut, the *Drosophila* midgut functions to absorb nutrients and acts as a barrier to pathogens (*Miguel-Aliaga et al., 2018*). The adult midgut consists of a single layer of polarised epithelial cells, with their apical sides facing the gut lumen and basal sides contacting the extracellular matrix (ECM) and muscle layer (*Baumann, 2001*; *Chen et al., 2018*; *Shanbhag and Tripathi, 2009*). The epithelium is predominantly composed of enterocytes (90%) and also contains entero-endocrine cells (ee) and their progenitors. The intestinal stem cells (ISCs) and their progeny, the enteroblasts reside on the basal side of the epithelium beneath the enterocytes (*Goulas et al., 2012*; *Micchelli and Perrimon, 2006*; *Ohlstein and Spradling, 2006*). Quiescent enteroblasts are morphologically indistinguishable from ISCs until they are activated to differentiate in response to damage or nutrient-dependent signals (*Nászai et al., 2015*; *O'Brien et al., 2011*; *Rojas Villa et al., 2019*). Over the past 15 years, multiple signaling pathways have been found to control ISC proliferation and differentiation, but much less is known about how differentiating enteroblasts insert into the midgut epithelium and polarize to form an apical domain (*Antonello et al., 2015*; *Chen et al., 2016*; *He et al., 2018*; *Jiang and Edgar, 2011*; *Sasaki et al., 2021*; *Wang and Hou, 2010*; *Wu et al., 2021*).

The adult midgut epithelium differs from all other *Drosophila* epithelia in several key respects. Firstly, it is a secondary epithelium that is derived from the endoderm, in contrast to most *Drosophila* epithelia which are directly descend from the cellular blastoderm epithelium (*Pitsidianaki et al., 2021*; *Tepass and Hartenstein, 1994a*; *Yarnitzky and Volk, 1995*). Secondly, the apical-basal polarity of the epithelium does not require the conserved epithelial polarity factors that polarize all ectodermal and mesodermal epithelia, although most of these factors are expressed in the midgut. Instead, the polarity of the midgut enterocytes depends on the interaction between the integrin adhesion complex and the basement membrane (*Chen et al., 2018*). Thirdly, the midgut cells polarise in a basal to apical direction as enteroblasts differentiate into enterocytes as they integrate from the basal side. By contrast, the other well-characterised *Drosophila* epithelia, such as the cellular blastoderm and the follicular epithelium form in an apical to basal direction (*Franz and Riechmann, 2010*; *Müller, 2018*). Finally, the midgut epithelium has an inverted arrangement of lateral junctions to other *Drosophila* epithelia. The occluding, smooth septate junctions, form at the top of the lateral domain above typical adherens junctions, containing E-cadherin (Ecad), α-catenin and Armadillo (β-catenin; *Baumann, 2001*; *Lane and Skaer, 1980*; *Tepass and Hartenstein, 1994a*; *Tepass and Hartenstein, 1994b*).

The smooth septate junctions are composed of the transmembrane proteins Mesh, Snakeskin (Ssk), Tsp2a and Hoka and the scaffolding protein, Coracle (Cora), and provide the barrier to paracellular diffusion (*Furuse and Izumi, 2017*; *Izumi et al., 2021*; *Izumi et al., 2016*; *Izumi et al., 2012*). The ISCs and early enteroblasts lie below the septate junctions between the enterocytes and only form adherens junctions with neighbouring cells, while mature enterocytes form both adherens junctions and septate junctions, as well as specialised tri-cellular junctions at the vertex between three cells. As an enteroblast differentiates, it therefore needs to develop new septate junctions with the neighbouring enterocytes as it inserts into the epithelium. This also means that the existing septate junctions between neighbouring enterocytes must be broken to allow the new cell to integrate. Furthermore, the barrier function of the epithelium must be maintained during this process to prevent the contents of the gut lumen, such as digestive enzymes and potential pathogens, from leaking into the body.

As the enteroblasts integrate into the epithelium and become enterocytes, they form a new apical domain with its characteristic brush border. The formation of an apical domain de novo has previously been studied in two other contexts. In the *Xenopus* embryonic epidermis and *Drosophila* renal tubules, progenitor cells and Stellate cells integrate into the outer layer through a process of a fast radial intercalation, in which they form the apical junction and apical domain at the same time. The apical domain is then specified after the cells have reached the lumen (*Campbell et al., 2010*; *Sedzinski et al., 2016*). A different process occurs during apical lumen formation in MDCK cell cysts in 3D culture. After the first division, the new membrane between the two daughter cells forms a structure called the AMIS (apical membrane initiation site), where apical proteins and fluids are secreted to create the lumen (*Bryant et al., 2010*). In most cells, AMIS formation depends on components of the midbody that forms during cytokinesis and on the development of tight junctions at the lateral margins of the cell contact region (*Blasky et al., 2015*; *Mangan et al., 2016*). The site of apical domain formation in inserting enteroblasts cannot be induced by the midbody, however, since the ISCs divide beneath the epithelial cell layer and the enteroblasts are post-mitotic when they integrate.

Here we characterize the process of enteroblasts integration into the midgut epithelium. Our analysis reveals that integrating enteroblasts generate an 'apical' domain with a brush border before they reach the gut lumen and have a free apical surface. This pre-assembled apical compartment (PAC) forms below the septate junction between the neighbouring enterocytes and contains all apical markers tested. Differentiating enteroblasts also form new septate junctions with the neighbouring enterocytes below the existing septate junction between these enterocytes. The new septate junctions then move apically as the cell breaks through the existing septate junctions to emerge on the apical surface with a fully formed apical brush border. By contrast, most enteroblasts lacking septate junction components fail to integrate and either form no apical domain or form an internal apical domain.

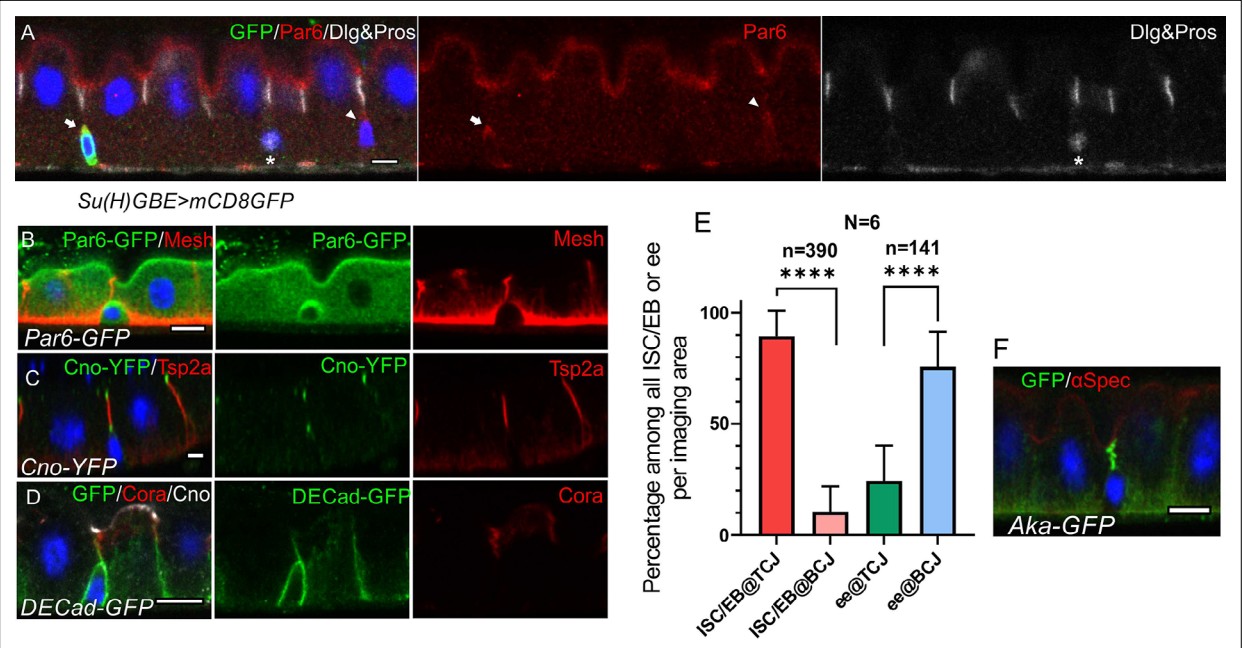

**Figure 1.** ISCs/early enteroblasts are polarised and reside underneath tri-cellular junctions. (**A**) Par-6 localises apically in ISCs and early enteroblasts. Su(H)GBE >mCD8 GFP expression (arrow) marks an early enteroblast, while the GFP-negative cell (arrowhead) is an ISC. Nuclear Prospero staining (white) marks an entero-endocrine cell (*) and cytoplasmic Dlg (greyscale) labels the septate junctions. (**B**) An ISC/early enteroblast expressing Par-6-GFP (green), which localises apically. Mesh (red) marks the septate junctions and basal labyrinth. (**C**) Canoe-YFP (green) localises to the marginal zone above the septate junctions (Tsp2a; red) in mature enterocytes. Although ISC/early enteroblasts do not form septate junctions, Canoe-YFP localises apically. (**D**) Adherens junctions (DE-cadherin-GFP; green) form throughout the cell-cell contacts between ISC/early enteroblasts and enterocytes. Coracle (red) marks the septate junctions; Canoe; white. (**E**) A graph showing the proportion of ISCs/early enteroblasts (ISC/EB) or entero-endocrine cells (ee) that localise beneath tri-cellular junctions (@TCJ) and bicellular junctions (@BCJ). Most ISCs/EBs localise beneath tricellular junctions where three enterocytes meet, whereas entero-endocrine cells mainly lie beneath bicellular junctions. DE-Cad-GFP expressing midguts were stained for GFP to mark cell contacts and Prospero to mark the ees. Cells with a low DNA content (~2 n) were counted by imaging from the basal side. 390 ISC/early enteroblasts and 141 ee were scored in 6 guts. (**F**) An example of an ISC/EB beneath a tricellular junction marked by Anakonda-GFP (Aka-GFP; green). Scale bars = 5 µm.

The online version of this article includes the following source data and figure supplement(s) for figure 1:

**Source data 1.** Source data for the graph as *Figure 1E*.

**Figure supplement 1.** Most ISCs and enteroblasts lie beneath tricellular junctions.

## Results

### ISCs/early enteroblasts are polarised and reside under tri-cellular junctions

To analyse the steps in enteroblast integration, we first characterized the polarity and arrangement of junctions in quiescent enteroblasts and ISCs, which lie on the basal side of the epithelium. At steady state, ISCs, quiescent enteroblasts and entero-endocrine cells are diploid (2 c/4 c)(*Rojas Villa et al., 2019*; *Xiang et al., 2017*). ISCs and early enteroblasts can therefore be identified by their low nuclear volume and lack of expression of Prospero, which specifically labels entero-endocrine cells (*Micchelli and Perrimon, 2006*; *Ohlstein and Spradling, 2006*). Both cell-types are already polarised, as they localise the apical polarity factor Par-6 in a crescent at the apex of the cell (*Figure 1A–B*, *Figure 1—figure supplement 1C*). Par-6 is not required for enterocyte polarity, but nonetheless provides a useful marker for the early polarization (*Chen et al., 2018*). The actin cytoskeleton is not polarised at this stage, however, and adherens junctions with the neighbouring enterocytes localize uniformly around the cell (*Figure 1D*, *Figure 1—figure supplement 1A and D*).

Canoe is a scaffolding protein that often links the actin cytoskeleton with junctional complexes (*Boettner et al., 2003*; *Perez-Vale et al., 2021*; *Sawyer et al., 2009*; *Yu and Zallen, 2020*). Canoe localises to the most apical tip of the septate junctions in differentiated enterocytes, forming a thin

belt separating the apical brush border (BB) from the lateral junctions (*Figure 1C–D*). This location is equivalent to the marginal zone in other fly epithelia and the vertebrate marginal zone (VMZ) in mammalian epithelial cells (*Tan et al., 2020*; *Tepass, 2012*). Like Par-6, Canoe also localises apically in ISCs and early enteroblasts, which have no septate junctions (*Figure 1B–D*, *Figure 1—figure supplement 1B-C*). The apical localisation of Canoe and Par6 in ISC/early enteroblasts indicates that they have some apical-basal polarity.

The basal ISCs are found as single cells or as pairs of cells, which can either be two ISCs or an ISC and an enteroblast (*de Navascués et al., 2012*; *Zhai et al., 2017*). Imaging ECad-GFP flies from the basal side reveals that the ISC/early enteroblasts (*Figure 1—figure supplement 1A*) usually lie underneath the tri-cellular junctions (TCJ) between enterocytes rather than bi-cellular junctions (BCJ) (*Figure 1E*). In a side view of the epithelial layer, the ISCs/early enteroblasts sit immediately beneath the TCJs, marked by Anakonda-GFP (*Figure 1F*). This means that enteroblasts need to break through three mature enterocyte-enterocyte septate junctions to integrate into the epithelial layer and gain access to the gut lumen. More importantly, this integration process needs to be tightly regulated to ensure that gut barrier function is maintained.

## Integrating enteroblasts generate a pre-assembled apical compartment (PAC)

Early enteroblasts express the Notch signalling reporter, Su(H)GBE-Gal4 >mCD8 GFP, but this is turned off as they differentiate (*Micchelli and Perrimon, 2006*; *Ohlstein and Spradling, 2007*; *Rojas Villa et al., 2019*). Enteroblasts that have just started to differentiate can therefore be identified as cells with larger nuclei, in which some GFP signal still perdures. The frequency of such early differentiating enteroblasts is very low under homeostatic conditions (*Reiff et al., 2019*). We therefore exposed the flies to a 2 hr heat shock at 37 °C to induce minor damage and dissected them 1 day later, which increases the number of dividing ISCs and differentiating Su(H)GBE >mCD8 GFP positive enteroblasts with larger nuclear volumes (*Figure 2—figure supplement 1A-B*). This treatment causes a small increase in Tor kinase activity at the very anterior and posterior ends of the midgut as shown by phospho-4E-BP1 staining, but does not increase pERK staining, an indicator of EGFR/MAPK signaling (*Figure 2—figure supplement 1C-D*). Thus, the heat shock induces sufficient stress to trigger ISC divisions and enteroblast activation/differentiation but does not activate a regeneration process.

A proportion of the Su(H)GBE >GFP positive, differentiating enteroblasts and the older GFP-negative enteroblasts form a "bubble" like structure inside the epithelial layer (*Figure 2A*). This structure is surrounded by membranes, as shown by the localisation of the membrane associated protein, α-Spectrin (*Figure 2A*). The actin marker sqh::Utrophin actin binding domain-GFP (sqh::Utr-ABD-GFP), which labels the brush border of mature enterocytes, also localises to the lower portion of the "bubble" (*Figure 1E*). Furthermore, the actin-rich portion of the 'bubble' is labeled by all other apical domain markers that we have examined, including Par-6, Myosin IA, Myosin 7a, the actin cross-linking proteins, Fimbrin and Cheerio (Lcp1/Pls3 and Filamin in mammals), Rab11, a marker for the apical recycling endosome and Picot, a transmembrane protein with homology to anion co-transporter proteins (*Figure 2B–E*, *Figure 2—figure supplement 1E and G*). This strongly suggests that the lower portion of the 'bubble' corresponds to a nascent apical domain. In support of this view, EM images reveal that this region forms a brush border (*Figure 2F*). Thus, the lower part of the 'bubble' structure has all the features of a pre-assembled apical domain, although it is not at the apical side of the epithelium.

To test whether the lumen facing the internal apical domain is continuous with the gut lumen, we examined the position of the brush border (sqh::UtrABD-GFP), relative to the septate junctions (Cora) and the marginal zone (Canoe). In most cases, the internal apical domain lay beneath the septate junction connecting the overlying enterocytes, indicating that the lumen lies within the epithelium and does not connect to the gut lumen (*Figure 2G and I*, *Figure 2—figure supplement 1H*, *Figure 2—video 1*). In other cases, the only septate junctions are the new junctions between the invaginating enteroblast and the neighbouring enterocytes and the lumen is continuous with the gut lumen (*Figure 2H and I*, *Figure 2—figure supplement 1I*, *Figure 2—video 2*). Thus, the enteroblasts first form an apical domain and lumen within the epithelium. The septate junction between the neighbouring enterocytes then disappears, allowing the internal lumen to fuse with the gut lumen and the apical domain to reach the apical surface. Since these results indicate that the apical domain

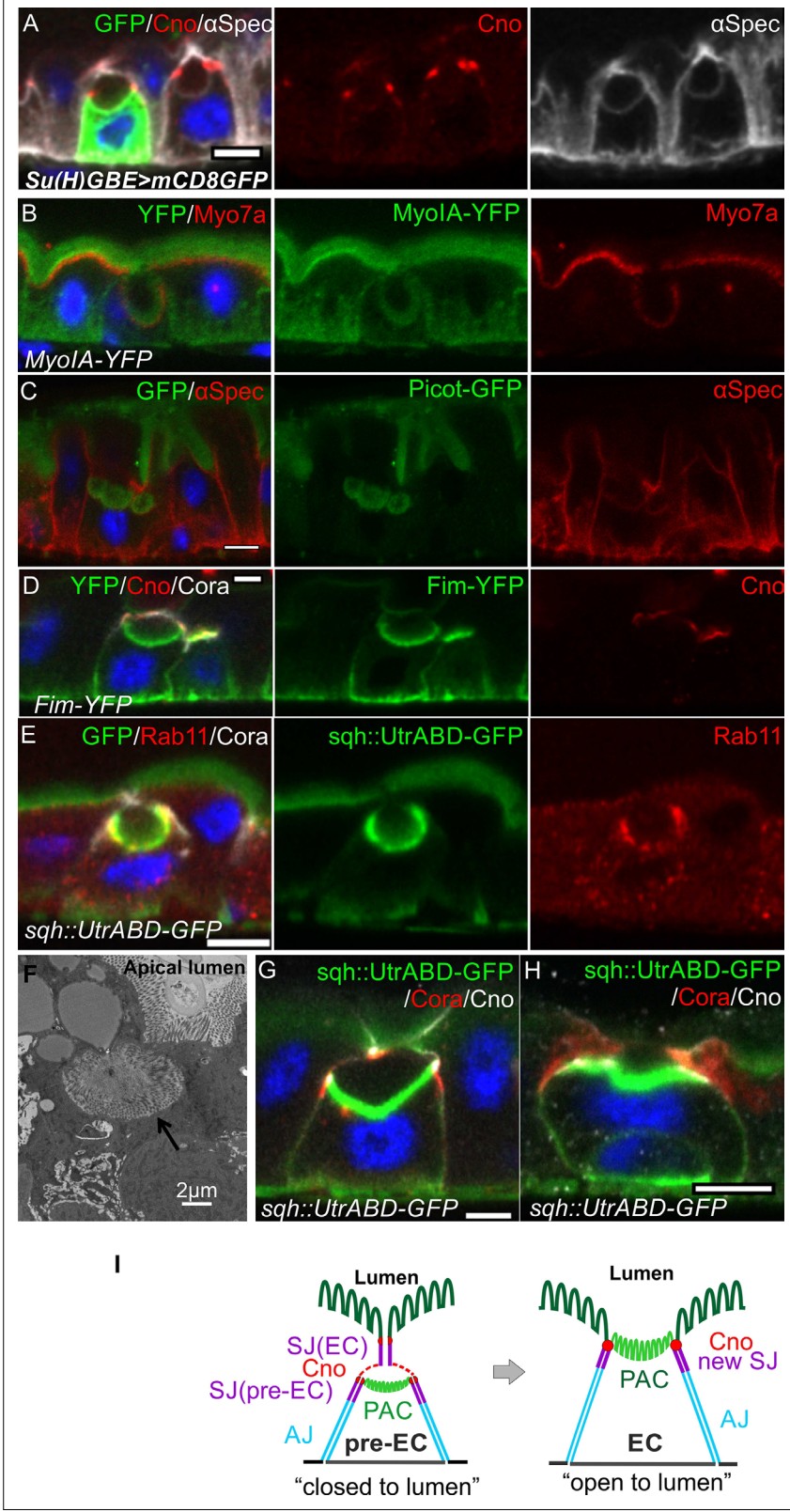

**Figure 2.** Integrating enteroblasts form an apical domain before reaching the gut lumen. (**A**) A transverse section of a *Su(H)GBE >mCD8* GFP midgut imaged 1 day after heat shock. Large spherical lumens surrounded by plasma membranes (α-spectrin; greyscale) have formed between the integrating enteroblasts and the overlying enterocytes. One enteroblast is still GFP-positive, indicating that it has recently been activated, whereas the other

*Figure 2 continued on next page*

*Figure 2 continued*

has lost Su(H)GBE >mCD8 GFP (green) expression. Canoe (red) labels the apical corners of the enteroblasts. (**B**) The lumen-facing side of the enteroblast is marked by MyoIA-YFP (green) and Myo7a (red), which are markers for the enterocyte brush border and apical cortex respectively. (**C**) A GFP protein trap line in the transmembrane transporter Picot (green) labels the apical brush border in enterocytes and the lumen-facing membrane in an integrating enteroblast. Note that multiple lumens have formed between the enteroblast and the enterocytes above. Plasma membranes are labelled with α-spectrin (red). (**D**) Fimbrin-GFP (Fim-GFP; green) marks the apical cortex of an integrating enteroblast and the enterocyte brush border. Note that the enteroblast to the right has not yet formed a lumen but has Fimbrin, Canoe (red) and Coracle (greyscale) localised to its apical surface. (**E**) The actin binding domain of Utrophin (Sqh::UtrABD-GFP; green) marks the enterocyte brush border and the apical side of an integrating enteroblast. The apical recycling endosome marker, Rab11 (red) also labels the apical region of the enteroblast. (**F**) A transmission electron micrograph showing that the lumen above an integrating enteroblast is surrounded by brush border microvilli. Scale bar, 2 μm. (**G**) An integrating enteroblast with a closed pre-apical compartment (PAC) and lumen that lie below the septate junction between the overlying enterocytes. The cells express the actin marker, Sqh::UtrABD-GFP (green), and are stained for Coracle (red) and Canoe (greyscale). (**H**) An integrating enteroblast stained as in (**H**) with an open lumen that is continuous with the gut lumen. (**I**) A model for enteroblast integration in which a 'closed' lumen above the PAC precedes an 'open' lumen. The 'closed' lumen stage represents the pre-EC with a PAC forming underneath the septate junction between the neighbouring enterocytes (purple), creating an isolated, closed lumen inside the epithelial layer. The cap over the 'closed' lumen comes from the neighbouring enterocytes. Adherens junctions form between pre-EC (light blue) and neighbouring enterocytes. New septate junctions also form between the pre-EC and the adjacent enterocytes (purple). The 'open' lumen represents a fully-developed enterocyte after the lumen has fused with the gut lumen, turning the PAC into the apical domain. To simplify the cartoons in the following figures, we combine the membranes between pre-EC and neighbouring ECs into one line. Scale bars in A-E, G and H, 5 μm.

The online version of this article includes the following video, source data, and figure supplement(s) for figure 2:

**Figure supplement 1.** Heatshock increases the number of integrating enteroblasts without inducing a regeneration response.

**Figure supplement 1—source data 1.** Source data for graph as *Figure 2—figure supplement 1A*.

**Figure supplement 1—source data 2.** Source data for graph as *Figure 2—figure supplement 1B*.

**Figure 2—video 1.** Stack images for Figure 2G, showing the 'closed' lumen above the PAC.
https://elifesciences.org/articles/76366/figures#fig2video1

**Figure 2—video 2.** Stack images for Figure 2H, showing the 'open' lumen with a new enterocyte exposed to the gut lumen, it has a concave shape apical domain which is originated from the PAC.
https://elifesciences.org/articles/76366/figures#fig2video2

---

forms internally, we refer to this structure as a 'pre-assembled Apical Compartment (PAC)' and refer to the cells with a PAC as pre-enterocytes (*Figure 2I*). In the 'closed' PAC stage, when the PAC is not exposed to the gut lumen, a new set of septate junctions form beneath the septate junction between the neighbouring enterocytes. This means that the pre-enterocyte has septate junctions with its neighbours and a fully-developed apical brush border before it emerges on the apical surface, thereby maintaining the gut barrier during enteroblast integration (*Figure 2I*).

## Integrating enteroblasts initiate PAC formation by forming an apical AMIS

Imaging heat shocked flies expressing *sqh::UtrABD-GFP* or *Fim-GFP* and stained for Canoe and Cora revealed three distinct stages of PAC formation. Based on cell size and marker protein localisation, we infer the following sequence of steps. In the first stage, actin is diffusely distributed around the cell cortex, as it is in quiescent enteroblasts, but Canoe and Cora enriched apically (*Figure 3A*, *Figure 3—figure supplement 1A and C*). Actin then becomes enriched apically in a slightly smaller domain than Canoe and Cora (*Figure 3B*, *Figure 3—figure supplement 1B and D*). This co-localisation of actin and junctional proteins is reminiscent of the apical membrane initiation site (AMIS) in MDCK cells (*Bryant et al., 2010*). Coincident with the apical enrichment of actin, a slight separation appears between the apical membrane of the enteroblast and the overlying enterocyte membranes, suggesting that fluid is being secreted into this space (*Figure 3B*). The actin staining then becomes concentrated in the centre of the apical domain, while Canoe and Cora are depleted from this region, creating a central actin-rich zone that lacks junctional proteins (*Figure 3C*). This small actin region then expands, bends

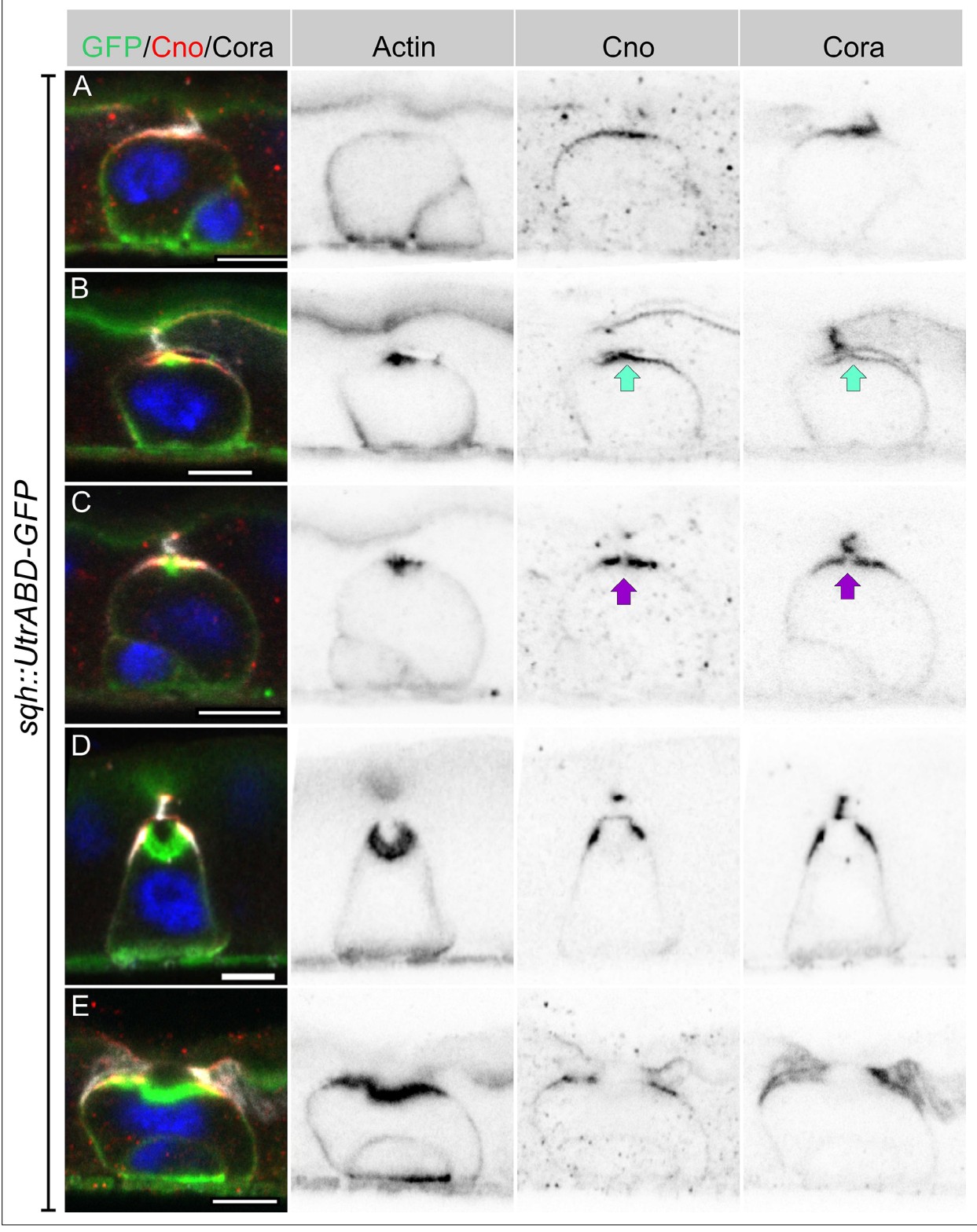

**Figure 3.** Integrating enteroblasts form an AMIS before forming a PAC. (**A**) When enteroblasts reach the level of the septate junction between the adjacent ECs, Canoe (red) and Cora (greyscale) localise to the apical side of the enteroblast, whereas actin is still uniformly distributed around the enteroblast cortex (green, GFP staining in *sqh::UtrABD-GFP* flies). (**B**) AMIS (green arrow) formation in an integrating enteroblast. Actin now localises to the apical side of the enteroblast in a smaller domain than Cora and Canoe. Note that the plasma membrane in the AMIS forming region has detached from the neighbouring ECs. (**C**) Actin concentrates in the centre of the AMIS, while Canoe and Cora are excluded from this region (purple arrow).

*Figure 3 continued on next page*

Figure 3 continued

(D) The actin enrichment enlarges to form a pre-assembled apical compartment (PAC) below the septate junction between the neighbouring ECs. Cora localises to the septate junctions that are forming between the pre-enterocyte and mature enterocytes on either side. Canoe localises to the marginal zones above the septate junctions and is also enriched on the enterocyte membranes facing the PAC lumen. (E) The septate junction between the overlying enterocytes dissolves and allows the closed lumen to fuse with the gut lumen. Scale bars = 5 µm.

The online version of this article includes the following video and figure supplement(s) for figure 3:

**Figure supplement 1.** Actin and septate junction protein localisation during AMIS and PAC formation.

**Figure 3—video 1.** Stack images for Figure 3D, showing the 'closed' lumen above the PAC.

https://elifesciences.org/articles/76366/figures#fig3video1

inwards and becomes the PAC, whereas Cora localises to the newly-forming septate junctions on either side, with Canoe slightly more apical in the nascent marginal zone (*Figure 3D*, *Figure 3—figure supplement 1E-F*). Finally, the pre-existing septate junction above the PAC disappears, and the PAC everts to form the apical domain (*Figure 3E*).

## AJs are cleared from the AMIS

The separation between the enteroblast and enterocyte membranes as the AMIS forms suggests that cell-cell adhesion must be modified at this stage. We therefore examined the localisation of Ecad-GFP and Armadillo during enteroblast integration. Early enteroblasts that have not reached the septate junction between the overlying enterocytes form adherens junctions (AJ) with the adjacent enterocytes all round their contacting surfaces (*Figure 4A*). At this stage, Cora is uniformly distributed around the cortex and Canoe is weakly enriched apically. Canoe and Cora become polarised apically when the integrating enteroblast contacts the enterocyte-enterocyte septate junction above, but the adherens junctions remain uniformly distributed (*Figure 4B*). As actin and Canoe become enriched apically to form the AMIS, adherens junctions are lost from this region and the enteroblast membrane separates from the membranes of the overlying enterocytes (*Figure 4C*). Adherens junctions remain absent from this region as the PAC starts to form (*Figure 4D*, *Figure 4—figure supplement 1A-B*). Thus, AMIS formation is associated with the loss of adherens junctions. This is presumably required to allow the enteroblast membrane to separate from that of the overlying enterocytes, as fluid is secreted from the AMIS to form the internal lumen.

## Sox21a is turned off at the enteroblast to pre-enterocyte transition

The transcription factor Sox21a is required for the differentiation of enteroblasts into enterocytes and can induce the precocious differentiation of quiescent enteroblasts when over-expressed, in part by activating the expression of another transcription factor, Pdm1 (*Chen et al., 2016*; *Meng and Biteau, 2015*; *Zhai et al., 2017*). We therefore examined the expression of Sox21a during enteroblast integration and differentiation in heat-shocked flies expressing *sqh::UtrABD-GFP*. The Sox21a antiserum labels the septate junctions, but this signal is non-specific, as it is still present in *sox21a* mutant cells (*Figure 5—figure supplement 1A*). Nuclear Sox21a is present at high levels in unpolarised enteroblasts, which probably correspond to the oval-shaped enteroblasts described by *Chen et al., 2016*, but the levels are significantly lower in polarised enteroblasts (*Figure 5A, B and D*). Sox21a is no longer detectable above background in the nuclei of pre-enterocytes that have formed a PAC, with a nuclear intensity similar to that in the neighbouring Sox21a-negative enterocytes (*Figure 5B and D*). Indeed, the pre-enterocytes are similar to enterocytes at the transcriptional level, as they also express the enterocyte marker, Pdm1 (*Figure 5C*). Sox21a levels are therefore inversely correlated with enteroblast differentiation and integration: unpolarised enteroblasts have high Sox21a, polarised enteroblasts with an AMIS have lower levels, and pre-enterocytes with a PAC are Pdm1 +and Sox21a- like the neighbouring enterocytes (*Figure 5E*).

To estimate the time taken for enteroblasts to progress to pre-enterocytes with a PAC, and for pre-enterocytes become to enterocytes, we induced enterocyte differentiation by over-expressing *UAS-Sox21a* under the control of *esg[ts]-Gal4* and counted the number of GFP+ve cells without a PAC, with a PAC and with a full apical domain at different time points after induction (*Chen et al., 2016*; *Meng and Biteau, 2015*; *Zhai et al., 2017*). 17 hr after shifting the flies to 25 °C to inactivate Gal80ts, almost no GFP+ve cells had progressed to pre-EC with a PAC (0.1%) or EC (1%), and these few cells probably

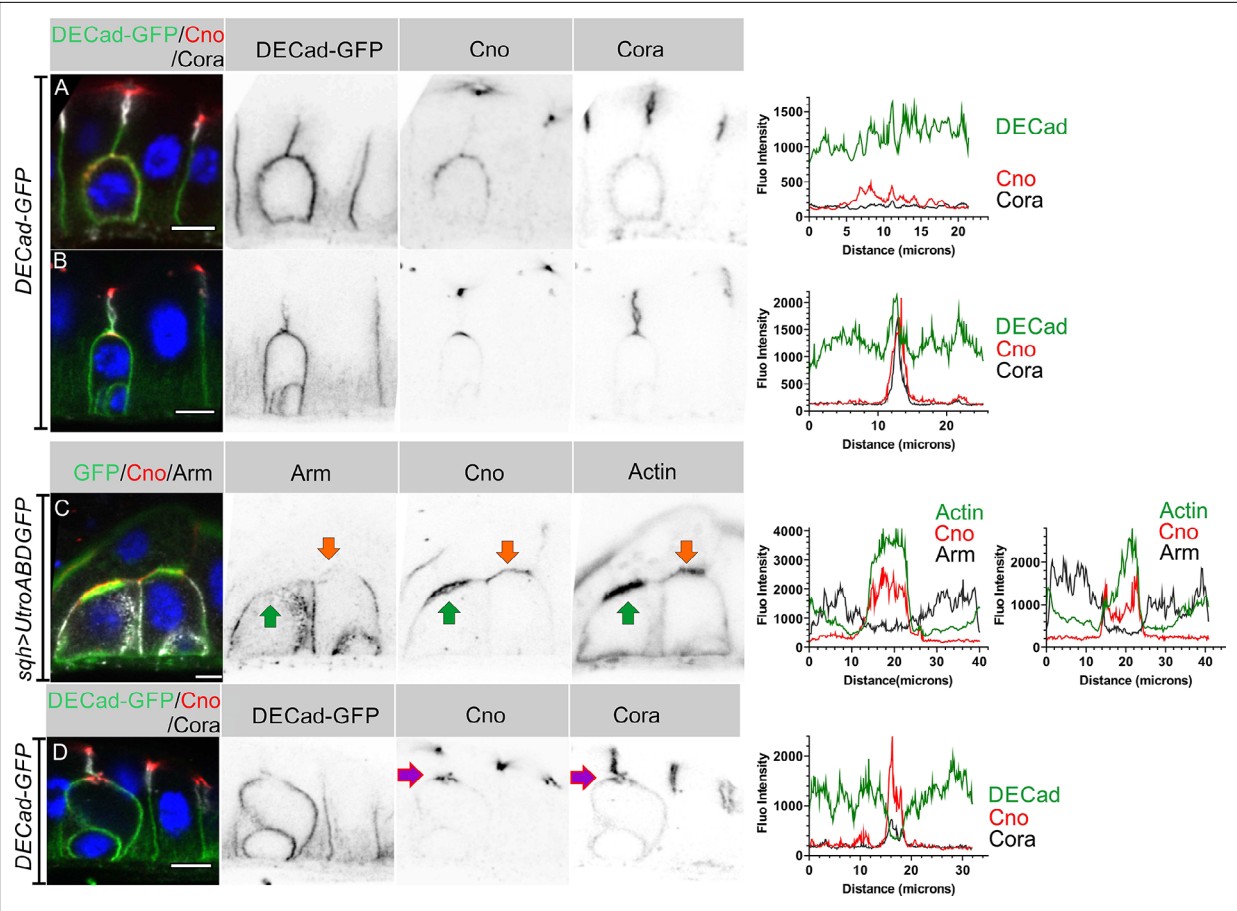

**Figure 4.** AJs are lost from the enteroblast apical membrane during AMIS formation. (**A**) E-cadherin (green) localises all around the plasma membrane of integrating enteroblasts that have not yet reached the septate junction between the overlying enterocytes. Cora (greyscale) is not polarised at this stage and Canoe (red) is only weakly enriched apically. (**B**) Canoe and Cora form apical caps in enteroblasts that have reached the septate junction between the overlying enterocytes, but E-cadherin is still uniformly distributed along the enterocyte-enteroblast cell contact sites. This enteroblast is at the stage when Canoe is polarised but actin is not (see *Figure 3A*, *Figure 3—figure supplement 1A and C*). (**C**) AMIS formation (green arrow) in an integrating enteroblast. The Adherens junctions (Arm staining in white) disappear from the AMIS region. Canoe (red) localises to both the enteroblast and enterocyte membranes after their separation as seen in *Figure 3B*. The integrating enteroblast on the right (orange arrow) is at a slightly earlier stage before separation of the apical enteroblast membrane from the overlying enterocyte membranes. Actin (green) has started to accumulate apically, but at lower levels than in the left-hand enteroblast. (**D**) Ecad-GFP (green) is absent from the membrane around the pre-assembled apical compartment (purple arrow) as it forms. The fluorescence intensities of all labelled components are plotted in the panels next to the corresponding figures. Scale bars = 5 µm.

The online version of this article includes the following source data and figure supplement(s) for figure 4:

**Source data 1.** Source data for the fluorescence intensity plot in *Figure 4A–D*.

**Figure supplement 1.** The localisation of junctional proteins during PAC formation.

started to differentiate before Sox 21 a induction. 24 hr later, 10% of the GFP[+ve] cells had developed into pre-ECs with a PAC and 20% had become ECs. After an additional 24 hr, the number of cells with a PAC fell to 1%, whereas 50% were ECs (*Figure 5—figure supplement 1B-C*). Assuming that it takes 12–17 hr to induce high levels of Sox21a expression, these results suggest that most activated EBs take about 24 hr to develop into a pre-EC with a PAC and a further 24 hr to differentiate into a mature EC, although some cells differentiate faster. This time frame is in agreement with a previous study using similar approaches to accelerate differentiation (*Rojas Villa et al., 2019*) and a recent live imaging study tracing the enteroblast to enterocyte transition (*Tang et al., 2021*). This experiment also indicates that down-regulation of Sox21a is not essential for enteroblast to pre-enterocyte

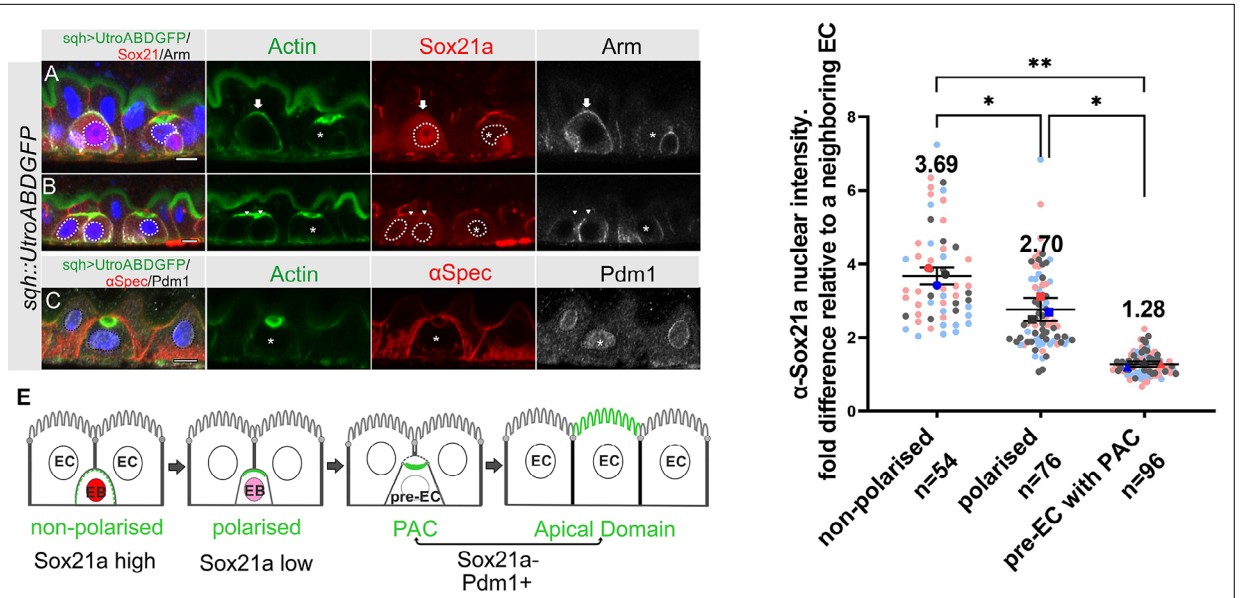

**Figure 5.** Sox21a levels fall as the PAC forms during integration. (**A–B**) Sox21a (red) is present at high levels in the nuclei of enteroblasts in which actin (Utr-ABD-GFP; green) is not yet polarised apically (arrow in A), is lower in the nuclei of enteroblasts with polarised actin (arrowheads in B). Pre-enterocytes that have formed a PAC (asterisks in A and B) lack nuclear Sox21a. The nuclei are outlined by white dashed lines. Note that the Adherens junctions (Armadillo; greyscale) still extend around the apical membrane of the enteroblast with unpolarised actin, but this signal has disappeared in the enteroblasts with apical actin (AMIS stage, arrowheads in B) and the pre-enterocytes with a PAC (*). The anti-Sox21a antiserum labels the septate junctions, but this is non-specific staining as it is still present in *Sox21a* mutant flies (see *Figure 5—figure supplement 1A*). (**C**) Pdm1+ (white) is expressed in a Pre-enterocyte with a PAC (asterisk); actin in green and αSpec in red. Scale bar = 5 μm. (**D**) Graph showing the levels of nuclear Sox21a staining relative to neighbouring enterocytes at different stages of enteroblast integration. The horizontal lines indicate the median values, which are significantly different by a two-tailed t test among three groups (**p<0.005, *p<0.05). n, the number of EB/neighbouring EC pairs. (**E**) Diagram showing the levels of nuclear Sox21a and Pdm1 during enteroblast integration.

The online version of this article includes the following source data and figure supplement(s) for figure 5:

**Source data 1.** Source data for the graph as *Figure 5D*.

**Figure supplement 1.** Activated enteroblasts take 1-2 days to become enterocytes.

**Figure supplement 1—source data 1.** Source data for the graph as *Figure 5—figure supplement 1C*.

differentiation, since enteroblasts overexpressing Sox21a still from a PAC (*Figure 5—figure supplement 1B*).

Combining all our analyses, we propose the following steps in enteroblast differentiation (*Figure 6A*). When an enteroblast is activated by high Sox21a expression, it starts to grow in size but loses its apical-basal polarity, with adherens junctions all over its contacting surface and uniform cortical actin. Once the enteroblast reaches the septate junction between the neighbouring enterocytes, Canoe and Cora become localised apically. Shortly afterwards, the apical membrane initiation site forms, with the apical localisation of F-actin and the removal of DE-cadherin from this region. This coincides with the down-regulation of Sox21a and the opening of a space between the plasma membranes of the enteroblast and the overlying enterocytes, suggesting that the apical membrane is secreting fluid between the cells. As the cell differentiates into a pre-enterocyte, F-actin segregates from Canoe and Coracle into the centre of the AMIS to form the nascent PAC. The PAC then expands and new septate junctions marked by Cora, Mesh (*Figure 2—figure supplement 1G*) and Tsp2A (*Figure 4—figure supplement 1A*) form between the enteroblast and the adjacent enterocytes, sealing the fluid-filled lumen above the PAC. This coincides with the localisation of Canoe to the marginal zone above the new septate junctions and in the enterocyte membrane covering the PAC. By this stage, the pre-enterocyte no longer expresses Sox21a and has activated Pdm1. Finally, the septate junctions above the internal lumen disassemble, allowing the membranes of the overlying enterocytes to separate, so that the lumen becomes continuous with the gut lumen. At this point the PAC becomes the apical domain of the newly-integrated enterocyte (*Figure 6A–B*). The mechanisms that drive the disassembly of the overlying septate junctions are not known, but it is worth bearing in

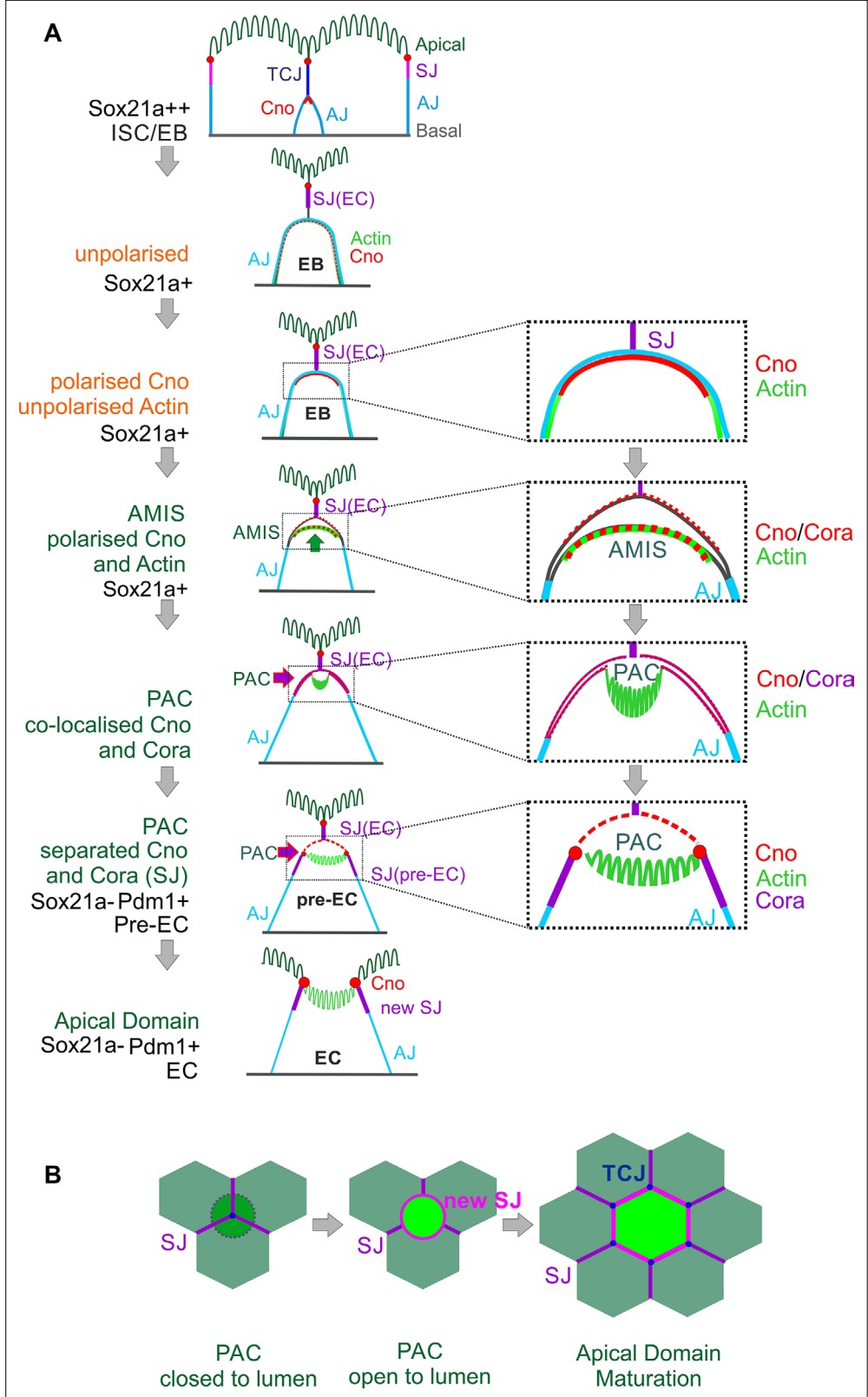

**Figure 6.** Diagram of the steps in enteroblast integration. (**A**) Diagram shows the side view of the steps in enteroblast integration. 'Unpolarised' in the second panel of this figure indicates that the enteroblast has not formed a distinct apical domain. At this stage, no marker is clearly apically localised. 'unpolarised' or 'polarised' in the third and fourth panels describe the localisation of marker proteins such as Actin and Canoe. (**B**) Top

*Figure 6 continued on next page*

*Figure 6 continued*

view model of the last steps from 'closed' (dashed line) to 'open' lumen, and to the stage when the new EC forms a mature apical domain (green) and builds new SJs (purple) and tri-cellular junction (TCJ) (navy blue) with neighbouring ECs.

mind that most EBs integrate beneath a tricellular junction. This tricellular junction must therefore also be disassembled to allow the new enterocyte to emerge on the apical surface, and three new tricellular junctions must eventually form where the new enterocyte meets two of the adjacent enterocytes. Indeed, the disassembly of the overlying tricellular junction may be the first event that triggers the emergence of the new enterocyte (*Figure 6B*).

## Neither Canoe or Coracle is required for PAC formation or cell polarity

Canoe provides a useful marker to follow enteroblast integration, as it is the first protein detected apically as the enteroblast polarises and it then labels the AMIS and the outer rim of the PAC. During AMIS and PAC formation, we noticed that Canoe also localises to the enterocyte membranes that separate from the apical membrane of the enteroblast and cover the internal lumen (orange arrow in *Figure 4C*, yellow arrow in *Figure 7A*). Since Canoe normally localises to the marginal zone above the septate junctions in enterocytes, this suggests that enterocytes respond to the presence of an invaginating enteroblast/pre-enterocyte by re-localising Canoe. To confirm that this is the case and to test whether Canoe plays a functional role in enteroblast integration, we generated positively marked clones homozygous for the null allele *canoe*[R10] using the MARCM system (*Lee and Luo, 2001*; *Sawyer et al., 2009*). When one of the adjacent enterocytes is mutant for *canoe*, Canoe is lost from the enterocyte membrane covering that side of the internal lumen, confirming that this signal normally comes from the enterocyte (*Figure 7B*). Consistent with this, when the pre-enterocyte is mutant for *canoe*, staining is lost from the bottom surface of the PAC (*Figure 7C*, *Figure 7—figure supplement 1A-B*). In both cases, however, the PAC forms normally, suggesting that Canoe is not required for enteroblast integration or polarisation. To rule out the possibility that the lack of a phenotype was due to residual function of the *canoe*[R10] mutant, we used CRISPR to generate a 14 bp deletion and premature stop codon in DIL domain of Canoe. MARCM clones of this allele, *canoe*[ic], lack Canoe staining, but are otherwise wild-type, confirming that Canoe is not required for enteroblast integration or enterocyte polarity (*Figure 7D–E*, *Figure 7—figure supplement 1C-D*).

Coracle is a peripheral septate junction protein whose localisation depends on the structural septate junction components such as Mesh/Ssk/Tsp2a (*Chen et al., 2018*; *Izumi et al., 2016*; *Izumi et al., 2012*). Cora antibody staining provides a clearer marker for the septate junctions than Mesh or Tsp2a antibody staining, because the latter also label the basal labyrinth (*Figure 3—figure supplement 1E-F*). To determine whether Cora is required for PAC formation or epithelial polarity in the adult midgut, we generated a null mutant allele with a premature stop codon in the FERM domain using CRISPR. Cells mutant for this allele, *cora*[ic], or a second *cora* null allele, *cora*[5], can form a PAC, septate junctions and a full apical domain, indicating that Cora is also not required for enteroblast integration or enterocyte polarity (*Figure 7F–G*, *Figure 7—figure supplement 1E-H*).

## Septate junctions are required for normal PAC formation

The smooth septate junctions of the *Drosophila* midgut are formed by several transmembrane proteins, including Mesh, Tsp2A and Snakeskin (Ssk), which are specifically expressed in endodermal tissues (*Furuse and Izumi, 2017*). The apical localisation of these proteins coincides with the formation of the AMIS, while PAC formation correlates with their lateral displacement, raising the question of whether they play role in these processes.

We examined the effects of loss of *mesh* by generating positively marked MARCM clones of *mesh*[f04955] and a new *mesh* null allele, *mesh*[R2], which we identified as a second hit on the *canoe*[R2] chromosome (*Figure 8A*; *Sawyer et al., 2009*). *mesh*[R2] and *mesh*[f04955] homozygous cells never integrate fully into the epithelial layer and remain below the septate junctions of the overlying enterocytes, presumably because they cannot make septate junctions of their own (*Figure 8A–C*). Because the mutant cells remain trapped beneath enterocyte-enterocyte septate junctions, they accumulate in the basal region of the epithelium, with new EBs derived from the same mutant ISC forming beneath them and reducing their contact with the basement membrane (*Figure 8A*).

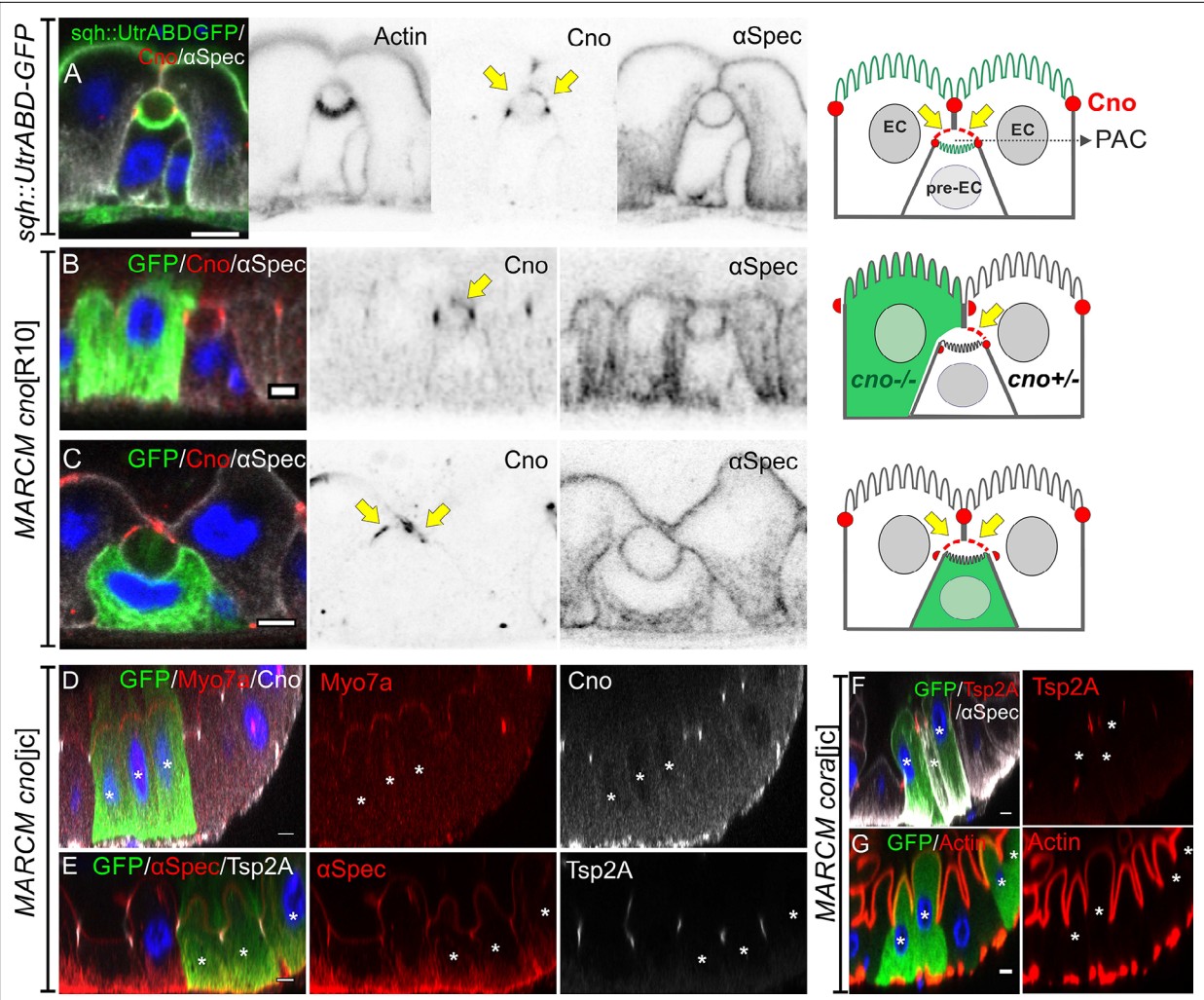

**Figure 7.** Neither Canoe or Coracle is required for PAC formation or enterocyte polarity. (**A**) Canoe (red) localises to the enterocyte membranes (yellow arrows) that face the lumen above an integrating pre-enterocyte, and to the marginal zone above the newly formed septate junctions between the pre-enterocyte and neighbouring enterocytes. Sqh >UtrABD GFP labelling of Actin in green and α-spectrin in greyscale. (**B**) An enteroblast integrating between a *canoe*[R10] mutant enterocyte (green) and a heterozygous enterocyte. Canoe (red) is lost from the roof of the lumen on the side with the mutant enterocyte, but still marks the roof on the side with a non-mutant enterocyte (yellow arrow). (**C**) A *canoe*[R10] mutant pre-enterocyte (green) integrating between two heterozygous enterocytes. The PAC still forms normally in the absence of Canoe (red). α-spectrin in greyscale. (**D–E**) MARCM clones of *canoe*[ic] homozygous mutant cells stained for Myo7a (red; **D**), α-spectrin (red; **E**) and Tsp2A (greyscale; **E**). The mutant cells form normal apical domains and septate junctions in the absence of Canoe. (**F–G**) MARCM clones of *coracle*[ic] homozygous mutant cells stained for Tsp2a (red in F), α-spectrin (greyscale in F) and Actin (red in G). The mutant cells form normal apical domains and septate junctions in the absence of Coracle. * marks the homozygous mutant cells. Scale bar = 5 μm.

The online version of this article includes the following figure supplement(s) for figure 7:

**Figure supplement 1.** Canoe is not required for PAC formation.

Some of the *mesh* mutant cells contain one or more circular, actin-rich structures that are labelled by α-Spectrin and the apical domain proteins, Par-6, Myo7a and Rab11 (*Figure 8C, E, G and H*). These structures resemble PACs, but do not face the external space between the enteroblast and enterocytes, instead forming inside the cells. Cells with an internal PAC account for 12% of the *mesh*[R2] enteroblasts. 81% of differentiating *mesh* mutant enteroblasts, based on their size and ploidy (>8 n), do not appear to be polarised, with actin and $\beta_H$-Spectrin diffusely distributed around the contacting cell surface (*Figure 8D and J*). The remaining mutant cells (7%) are polarised, with actin, Par-6 and Rab11 localised apically (*Figure 8E, F and J*).

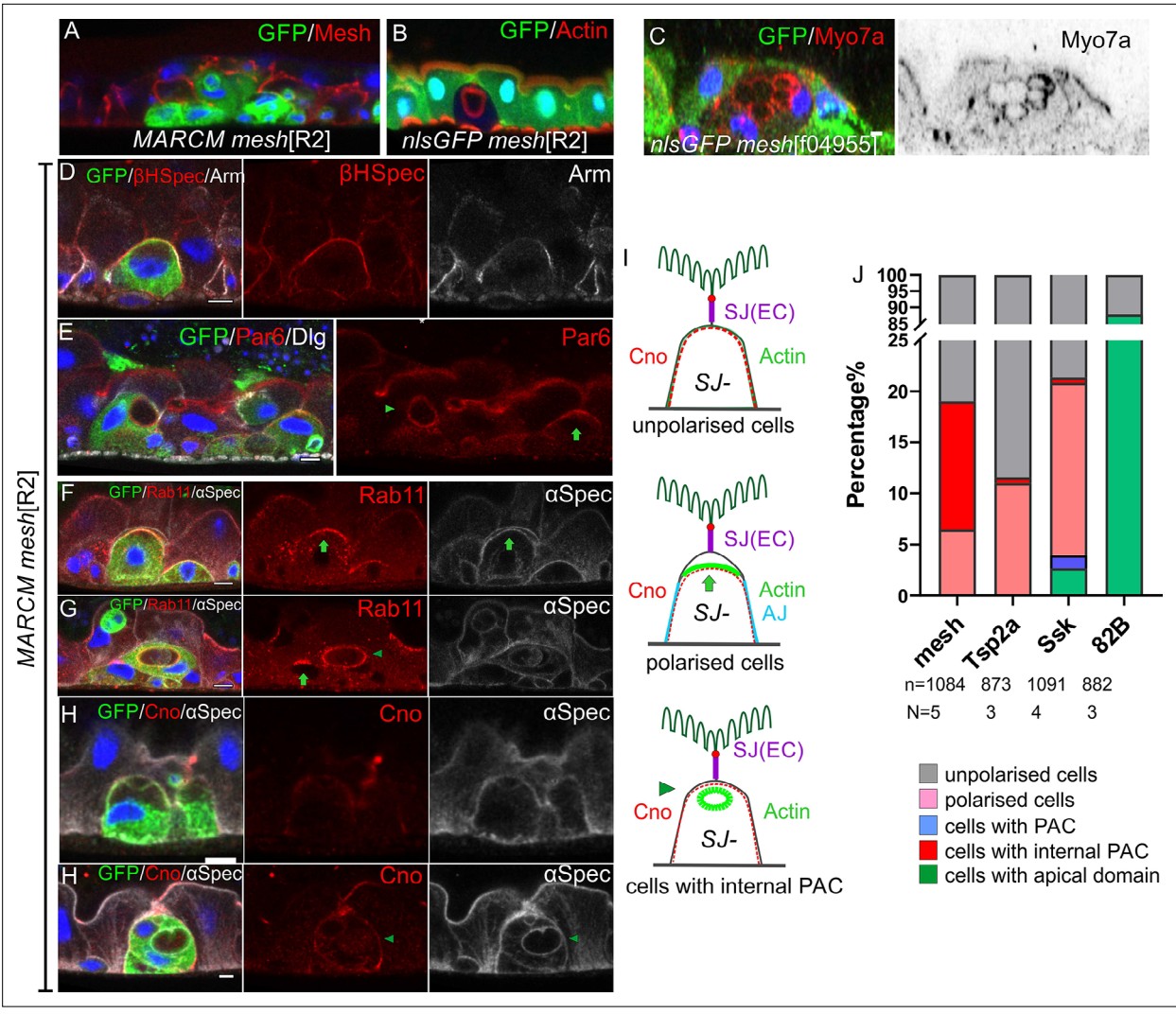

**Figure 8.** *mesh* mutants fail to form a PAC or form an internal PAC. (**A**) A *mesh*[R2] homozygous MARCM clone marked by GFP (green). The mutant cells lack Mesh staining (red), fail to make septate junctions and do not reach the gut lumen. (**B–C**) *mesh*[R2] (**B**) and *mesh*[f04955] (**C**) homozygous mutant cells marked by the loss of GFP (green). The mutant cells form internal PACs that are marked by actin (red in B) and Myo7a (red in C). The cell in C has formed multiple internal PACs. (**D**) $\beta_H$-spectrin (red) localises around the plasma membrane of *mesh*[R2] mutant enteroblasts (green), indicating that they do not polarise normally, but adherens junctions (Arm, greyscale) are still down-regulated at the apical surface. (**E**) *mesh*[R2] mutant enteroblasts (green) stained for Par-6 (red). Par-6 localises to the apical membrane (green arrow) in the enteroblast on the right, but localises around the internal PAC in the enteroblast on the left (green arrowhead). (**F**) A *mesh*[R2] mutant enteroblast stained for α-spectrin (greyscale) and Rab11 (red), which localises apically (green arrow). The α-spectrin staining reveals that a space has formed between the apical side of the integrating enteroblast and the neighbouring enterocytes. (**G**) A *mesh*[R2] mutant enteroblast stained for α-spectrin (greyscale) and Rab11 (red) that has formed an internal PAC. Rab11 decorates the surface of the internal PAC (green arrowhead). Note the younger, mutant enteroblast on the left (green arrow) localises Rab11 apically. (**H**) A *mesh*[R2] mutant enteroblast (green) with an internal PAC stained for Canoe (red) and α-spectrin (greyscale). Canoe does not localise to the internal PAC, which has no junctions. Scale bars = 5 μm. (**I**) Diagrams showing the distributions of Canoe, actin and adherens junctions in the three phenotypic classes of *mesh*[R2] mutant enteroblasts. The measurement of Canoe's intensity in the WT and *mesh*[R2] mutant enteroblasts is shown in *Figure 8—figure supplement 2* D&E. (**J**) Percentages of *mesh*, *Tsp2a* and *ssk* mutant enteroblasts in each of the classes in (**I**) compared to a wild-type control (FRT82B clones) based on actin in mutant cells (representative images in *Figure 8—figure supplement 1E and F*).

The online version of this article includes the following source data and figure supplement(s) for figure 8:

**Source data 1.** Source data for the graph as *Figure 8J*.

**Figure supplement 1.** Phenotypes of other septate junction mutants.

**Figure supplement 2.** *mesh* and *Tsp2a* mutants disrupt the apical localisation on Canoe in integrating enteroblasts.

**Figure supplement 2—source data 1.** Source data for the graphs as *Figure 8—figure supplement 2D and E*.

Mutants in other septate junction proteins also disrupt PAC formation, but cause a lower frequency of internal PACs. Only 0.6% *Tsp2a* mutant cells have an internal PAC and 11% still polarise, with the rest showing little or no sign of polarity (*Figure 8J*, *Figure 8—figure supplement 1E, F, K and L*). Similarly 0.5% of *ssk* mutant cells form an internal PAC, 17% polarise and 78.6% do not polarise (*Figure 8J*, *Figure 8—figure supplement 1H and I*). However, a small fraction of *ssk* mutant cells form a normal PAC (1.3%) and even integrate into the epithelial layer (2.6%) (*Figure 8J*, *Figure 8—figure supplement 1G and J*). By contrast, more than 80% of the cells in control *FRT82B* MARCM clones develop a mature apical domain, while the remaining ~10% lack apical actin and are presumably ISCs or early enteroblasts (*Figure 8J*). Mutants in each septate junction protein disrupt the localisation of the other septate junction components, as previously reported for the embryonic and larval gut (*Figure 8—figure supplement 1A-C*; *Izumi et al., 2021*; *Izumi et al., 2016*; *Izumi et al., 2012*). The differences between the spectrums of phenotypes observed in *mesh*, *Tsp2a* and *ssk* mutants therefore cannot be explained by their distinct effects on septate junction formation and may reflect differences in perdurance or other functions of these proteins, such as the regulation of the Yki pathway (*Chen et al., 2020*; *Izumi et al., 2019*; *Xu et al., 2019*).

The majority of cells mutant for septate junction components fail to polarise or form an AMIS, although they form normal lateral and basal domains, as shown by the normal localisation of the basal integrin signalling component, Talin (*Figure 8—figure supplement 1N*). This contrasts with the initial apical-basal polarisation of ISCs and quiescent enteroblasts, which are not affected in *mesh*, *Tsp2a* and *ssk* mutant clones (*Figure 8—figure supplement 2A-C*). The mutant clones also have similar sizes to the control empty FRT82B clones, indicating that they have no effect on stem cell behaviour or division rate (*Figure 8—figure supplement 1O*). Thus, the septate junction proteins are not involved in the initial polarisation of ISCs and quiescent enteroblasts, but they are required for the apical-basal polarisation that gives rise to the AMIS in most cells. This is distinct from the role of integrin adhesion complex, which is required for apical-basal polarity in ISCs, quiescent enteroblasts and activated enteroblasts (*Figure 8—figure supplement 2F-H*).

Some mutant enteroblasts still polarise to form an AMIS-like structure in the absence of septate junction proteins, since 5–20% of mutant cells still localise actin, Par-6 and Rab11 apically (Arrows in *Figure 8E–H*), lose their apical Adherens junctions from this region and detach their apical membrane from the overlying wild-type enterocytes (*Figure 8D and F*). In the case of *mesh* and *Tsp2A*, this AMIS-like structure never gives rise to a PAC. The low frequency of AMIS formation in these mutants may be due to partial rescue of the phenotype by the perdurance of the wild-type proteins after clone induction. Alternatively, the formation of the AMIS may not absolutely require the septate junction proteins, but only occur very inefficiently in their absence.

## Discussion

Enteroblasts perform an essential role in gut homeostasis by providing a pool of quiescent precursor cells that can rapidly integrate into the epithelium in response to damage or growth signals to replace or increase the number of enterocytes. In order to do this, the enteroblasts need to insert themselves between existing enterocytes, while maintaining the barrier function provided by the smooth septate junctions. Our results reveal that they do this by an unexpected mechanism, in which they form an apical domain inside the epithelial layer, immediately beneath the septate junction between the overlying enterocytes. At the same time, the enteroblast/pre-enterocytes form new septate junctions with their neighbours. This therefore provides an elegant solution to the problem of integrating into the epithelium without disrupting the barrier, by forming junctions between the enteroblast/pre-enterocyte and its neighbours before the septate junctions above the integrating cell are removed. This also means that the pre-enterocyte already has a fully differentiated apical domain with a brush border when it reaches the gut lumen, so that it can immediately assume its function in nutrient absorption. This sequence of events has been inferred from ordering images of many fixed samples based on cell size and the localisation of various proteins, but it will be important to corroborate this sequence by imaging enteroblast integration in living midguts. Nevertheless, our results are entirely consistent with another recent study of enteroblast integration in the adult midgut, which reached very similar conclusions (*Moreno-Roman et al., 2021*).

Quiescent enteroblasts are induced to differentiate in response to damage or growth signals from the visceral muscle, trachea, enterocytes and ISCs (*Jiang and Edgar, 2011*; *Miguel-Aliaga et al.,*

*2018*; *Nászai et al., 2015*). This leads to the down-regulation of *esg* and the Dronc caspase, which maintain enteroblasts in the undifferentiated state, and the up-regulation of factors that drive differentiation, such as Zfh2, Sox100b, Sox21a and Pdm1 (*Amcheslavsky et al., 2020*; *Jasper, 2020*; *Jiang et al., 2016*; *Meng et al., 2020*; *Meng and Biteau, 2015*; *Rojas Villa et al., 2019*). These changes induce the enteroblasts to increase in size under the control of the Insulin, TOR and EGFR signalling pathways and to endo-replicate and become polyploid (*Xiang et al., 2017*). Activated enteroblasts have also been found to develop basal, actin-rich protrusions and to migrate occasionally to other regions of the epithelium (*Antonello et al., 2015*; *Rojas Villa et al., 2019*). Unlike these earlier studies, which imaged the midgut from the basal side, we have analysed the apical-basal axis of the enteroblasts as they are activated. This reveals that growing enteroblasts go through a phase where they lack a distinct apical domain, in contrast to ISCs and quiescent enteroblasts, which are polarised and localise Canoe and Par-6 apically. These unpolarised enteroblasts are likely to correspond to the protrusive, migratory state of activated enteroblasts reported by Antonella et al., indicating that growing enteroblasts go through a mesenchymal stage (*Antonello et al., 2015*). Enteroblasts then re-polarise once they reach the septate junctions between the overlying enterocytes. Thus, the adult enteroblasts transiently lose polarity and become migratory before developing epithelial polarity, much like the embryonic midgut precursors, which undergo an epithelial to mesenchymal transition when they delaminate from the primary epithelium and become migratory, before re-polarising in contact with the visceral mesoderm to form the embryonic midgut epithelium (*Campbell et al., 2011*; *Pitsidianaki et al., 2021*; *Tepass and Hartenstein, 1994a*).

The re-polarisation of growing enteroblasts upon reaching the enterocyte-enterocyte septate junction occurs in two steps, with Canoe and septate junction proteins appearing at the apical membrane first, followed by F-actin. The apical localisation of actin coincides with the clearance of adherens junctions from the apical membrane and is followed by the appearance of a space between the apical membrane of the enteroblast and the overlying enterocytes. This suggests that the enteroblast is secreting fluid into the extracellular space between it and the enterocytes to separate the cell membranes, which are no longer held together via the E-cadherin adhesion. This may be driven by apical exocytosis of vesicles or the activity of water and ion channels. Actin and apical membrane markers such as Par-6, Picot, and Rab11 then start to concentrate in the centre of the AMIS, while septate junction proteins are excluded from this region and start to form new junctions with the neighbouring enterocytes laterally, thereby creating a seal on either side of the forming lumen. The central actin-rich region then expands to form the pre-apical compartment and often invaginates inwards, perhaps due to the fluid pressure of the lumen above.

The steps in the formation of the pre-assembled apical compartment bear several similarities with cyst formation in mammalian epithelial cultures in 3D, which also involves the development of an internal lumen at sites of cell-cell contact. In MDCK cysts, for example, the first sign of lumen development is the formation of an apical membrane initiation site, which is marked by the co-localisation of tight junction proteins, such as Cingulin and Occludin with the apical marker, Podocalyxin (*Bryant et al., 2010*). The AMIS then develops into a pre-apical patch as Cingulin and Occludin segregate away from the apical factors to form lateral tight junctions that seal the lumen, mirroring the behaviour of septate junction proteins in *Drosophila* enteroblasts (*Blasky et al., 2015*; *Mangan et al., 2016*). Similarly, the formation of an internal lumen in MDCK cysts is preceded by the loss of Cadherin from the contacting cell-cell surfaces, as it is in *Drosophila* enterocytes (*Ferrari et al., 2008*). Furthermore, junctional proteins are required to define the site of apical secretion in both systems, as knockdown of Cingulin blocks the formation of a single lumen in MDCK cysts and loss of Mesh or Tsp2a prevents the development of an external lumen in the *Drosophila* midgut (*Mangan et al., 2016*).

Despite these similarities, the spatial cues that determine where the AMIS forms appear to be different. In most mammalian epithelial cysts, the site of the AMIS is defined by the position of the midbody formed during the last cell division, whereas enteroblasts are postmitotic and are derived from intestinal stem cell divisions that may have occurred several days earlier (*Mangan et al., 2016*; *Rojas Villa et al., 2019*; *Tang et al., 2021*). The most likely cue for the repolarisation that creates the AMIS in *Drosophila* enteroblasts is the septate junction between the overlying enterocytes, as enteroblasts do not polarise until they reach this junction. 80–90% of enteroblasts mutant for septate junction components fail to polarise at this stage, suggesting that septate junction proteins play a role in sensing when the integrating enteroblast contacts the enterocyte-enterocyte septate junction.

However, this cannot be by forming three-way septate junctions with the existing septate junctions, because the enteroblast only forms septate junctions later and in a more lateral position.

Another important difference between the two systems is that mammalian epithelial cysts are symmetric, with the cells on both sides of the lumen forming apical domains, whereas only the integrating enteroblast/pre-enterocyte forms an apical domain in the fly midgut. The overlying enterocytes do not form a normal lateral domain over the lumen, however, as $\beta_H$-Spectrin and Canoe localise to the enterocyte cortex in this region, which is quite different to their localisations in mature enterocytes to the apical domain and the marginal zone above the septate junctions, respectively. This highlights the fact that the enterocytes above an integrating enteroblast are not passive bystanders but respond to the presence of the enteroblast and perhaps even also facilitate its integration. For example, the enterocytes must disassemble the septate junction above a pre-enterocyte for the latter to reach the gut lumen. This suggests that the enterocytes receive cues from the invaginating enteroblast that produce specific responses at each stage of the process, although the nature of these signals is not known.

The development of an enterocyte from an ISC involves first the loss of the apical-basal polarity shown by ISCs and quiescent enteroblasts and then the re-establishment of polarity as the AMIS forms. Mutants in components of the integrin adhesion complex, such as Talin and Kindlin, disrupt both the initial polarity seen in ISCs and quiescent enteroblasts and the repolarisation during AMIS formation. However, the formation of the AMIS and PAC also requires septate junction proteins. Thus, differentiating enteroblasts only require a basal cue to establish their initial apical-basal polarity, whereas the formation of the pre-assembled apical compartment also requires a junctional cue. The septate junctions are not necessary for apical domain formation per se, however, as some *mesh* mutant enteroblasts form a full-developed apical domain with a brush border inside the cell. This suggests that septate junctions define the site of apical domain formation by delimiting the region where apical membrane proteins are secreted to assemble the brush border, but do not control the process of apical domain formation directly. The internal apical domains in *mesh* mutant cells resemble the apicosomes observed in single human pluripotent stem cells in culture, which then fuse with the plasma membrane to form an extracellular lumen after cell division (*Taniguchi et al., 2017*). The apicosome is thought to form by the intracellular fusion of exocytic vesicles, and a similar mechanism may give rise to the internal apical domains in *mesh* mutant cells (*Taniguchi et al., 2017*). If this is the case, it suggests that the apical domain can self-assemble and the polarity system merely functions to position where this domain forms. We cannot rule out the alternative possibility, however, that the internal apical domains in *mesh* mutants form by endocytosis of the AMIS region, as has been proposed for the intracellular vesicles with brush borders observed in multi-villus inclusion disease (*Engevik et al., 2021*; *Engevik et al., 2019*).

The polarisation of the midgut epithelium does not require any of the canonical epithelial polarity factors that polarise all *Drosophila* epithelia derived from the ectoderm and mesoderm, and instead depends on basal cues from adhesion to the extracellular matrix (*Chen et al., 2018*). Our observations suggest that a possible reason for this difference is that integrating enteroblasts polarise in a basal to apical direction and form an apical membrane without having a free apical surface. This means that they cannot use the apical polarity cues that trigger the polarisation of all other *Drosophila* epithelia. Thus, the unusual polarity system in the midgut may be a consequence of the way the tissue is built and maintained by basal stem cells, and it will be interesting to determine if this is also the case for other epithelia with similar cellular arrangements.

## Materials and methods
### *Drosophila melanogaster* stocks
#### Fluorescently tagged protein lines
sqh::UtrABD-GFP (gift from Thomas Lecuit, The Developmental Biology Institute of Marseille (IBDM), France), Par-6-GFP (*Wirtz-Peitz et al., 2008*), MyoIA-YFP (*Lowe et al., 2014*) (Kyoto DGRC #115611), Canoe-YFP (*Lowe et al., 2014*) (Kyoto DGRC#115111), Fimbrin-GFP (Bloomington #51562), Cheerio-YFP (Kyoto DGRC#115514), Picot-GFP (Bloomington #50822), DECad-GFP (gift from Yang Hong, University of Pittsburgh, USA), Anakonda-GFP (*Byri et al., 2015*) (gift from Stefan Luschnig, University of Münster, Germany).

## Mutant stocks

FRT19A Tsp2A[1-2], Tsp2A[3-3], Tsp2A[2-9] (*Izumi et al., 2016*) (gifts from Mikio Furuse, National Institute for Physiological Sciences, Okazaki, Japan), FRT82B mesh[R2] (this paper), FRT82B mesh[f04955] (Kyoto DGRC #114660), FRT80B ssk[1] and ssk[2] (*Chen et al., 2020*) (gifts from Tony Ip, University of Massachusetts Medical School, Worchester, MA, USA), FRT82B canoe[R10] (*Sawyer et al., 2009*) (gift from Mark Peifer, University of North Carolina, Chapel Hill, NC, USA), FRT82B canoe[JC1] (this study), FRT2A Sox21a[6] (*Zhai et al., 2015*) (Bloomington #68389), FRT2A Fit1[18]Fit2[83], rhea[79a] (*Klapholz et al., 2015*) (gifts from B Klapholz and N Brown, Department of Physiology, Development and Neuroscience, University of Cambridge, UK),

UAS responder lines: UAS-Sox21a (*Zhai et al., 2015*) (Bloomington #35748),
Su(H)GBE >GFP: Su(H)GBE-Gal4, UAS-mCD8GFP
esg[ts]: esg-Gal4, UAS-mCD8GFP, tub >Gal80[ts]
Delta[ts]: Delta-Gal4, UAS-mCD8GFP, tub >Gal80[ts]

The following stocks were used to generate (positively labelled) MARCM clones (*Lee and Luo, 2001*):

MARCM FRT82B: y w, UAS-mCD8::GFP, Act5C-GAL4, hsFLP[1];; FRT82B tubP-GAL80.
MARCM FRT19A: w, hsFLP, tubP-GAL80, FRT19A;; tubP-GAL4, UAS-mCD8::GFP/TM3, Sb.
MARCM FRT2A: hsFLP[1]; tubP-GAL4, UAS-mCD8::GFP/CyO, GFP; FRT2A tubP-GAL80 (gift from B. Klapholz and N. Brown).
MARCM FRT80B: hsFlp[1]; tubP-GAL4, UAS-mCD8::GFP/CyO, GFP; FRT80B tubP-GAL80 (generated by Dr. Mihoko Tame)

Negatively marked *mesh* mutant clones were generated using the following stock: esg-GAL4, UAS-FLP, tubP-GAL80[ts]/CyO; FRT82B nlsGFP (gift from Dr. G. Kolahgar, Department of Physiology, Development and Neuroscience, University of Cambridge, UK).

## Stock maintenance

Standard procedures were used for *Drosophila* husbandry and experiments. Flies were reared on standard fly food supplemented with live yeast at 25 °C. For the conditional expression of UAS responder constructs (e.g., overexpression of Sox21a) in adult flies, parental flies were crossed at 18 °C and the resulting offspring reared at the same temperature until eclosion. Adult offspring were collected for 3 days and then transferred to 29 °C to inactivate the temperature sensitive GAL80[ts] protein. To generate MARCM or GFP-negative clones, flies were crossed at 25 °C and the resulting offspring were subjected to heat shocks either as larvae (from L2 until eclosion) or as adults (5–9 days after eclosion). To generate homozygous mutant clones of *mesh*, *ssk,* and *Tsp2a*, flies were heat-shocked as larvae (from L2 until eclosion) and dissected 4 days after eclosion. Heat shocks were performed at 37 °C for 1 hr twice daily. Flies were transferred to fresh food vials every 2–3 days and kept at 25 °C for at least 4 days after the last heat shock to ensure that all wild-type gene products from the heterozygous progenitor cells had turned over. All samples used in this study were obtained from adult female flies.

## Heat shock treatment in adult flies

Flies were subjected to heat shock in a 37 °C incubator for 2 hr in a horizontal vial containing fly food but not live yeast. After removal from the incubator, live yeast was added to the vial and the flies were kept at 25 °C for 24 hr before dissection.

To test the effect of heat shock treatment on ISC divisions, we counted the number of pH3[+] cells in attp2 (Bloomington #25710) flies as control. Enteroblast nuclear volume was measured using *Su(H)>mCD8* GFP flies using Fiji. Superplots were made using GraphPad Prism 9 software (*Lord et al., 2020*).

## Formaldehyde fixation and heat fixation

Detailed procedure was described previously (*Chen et al., 2018*; *Chen and St Johnston, 2022*). Samples were dissected in PBS and fixed with 8% formaldehyde (in PBS containing 0.1% Triton X-100) for 10 min at room temperature. Following several washes with PBS supplemented with 0.1% Triton X-100 (washing buffer), samples were incubated in PBS containing 3% normal goat serum (NGS, Stratech Scientific Ltd, Cat. #005-000-121; concentration of stock solution: 10 mg/ml) and 0.1% Triton

X-100 (blocking buffer) for 30 min. This fixation method was only used for samples in which F-actin was stained with fluorescently labelled phalloidin, as phalloidin staining is incompatible with heat fixation. The heat fixation protocol is based on a heat–methanol fixation method used for *Drosophila* embryos (*Müller, 2008*). Samples were dissected in PBS, transferred to a wire mesh basket, and fixed in hot 1 X TSS buffer (0.03% Triton X-100, 4 g/L NaCl; 95 °C) for 3 s before being transferred to ice-cold 1 X TSS buffer and chilled for at least 1 min. Subsequently, samples were transferred to washing buffer and processed for immunofluorescence stainings.

## Antibody stainings

Antibody stainings were performed as described previously (*Chen et al., 2018*; *Chen and St Johnston, 2022*). After blocking in blocking buffer (1xPBS, 0.1%TritonX-100, 5%NGS (vol/vol)), samples were incubated with the appropriate primary antibody/antibodies diluted in blocking buffer at 4 °C overnight. Following several washes in washing buffer (1xPBS, 0.1%TritonX-100), samples were incubated with the appropriate secondary antibody/antibodies either at room temperature for 2 hr or at 4 °C overnight. Samples were then washed several times in washing buffer and mounted in Vectashield containing DAPI (Vector Laboratories) on borosilicate glass slides (No. 1.5, VWR International). All antibodies used in this study were tested for specificity using clonal analysis (MARCM) or RNAi.

### Primary antibodies

Mouse monoclonal antibodies: anti-Dlg (4F3), anti-Cora (c615.16), anti-αSpec (3A9), anti-Arm (N2 7A1), anti-Pros (MR1A), anti-Talin (A22A). All monoclonal antibodies were obtained from the Developmental Studies Hybridoma Bank and used at a 1:100 dilution.

Rabbit polyclonal antibodies: anti-p4EBP1 (Phospho-4E-BP1 (Thr37/46) (236B4) Rabbit mAb #2855#2855T, Cell Signaling Technology), anti-pERK (Phospho-p44/42 MAPK (Erk1/2) (Thr202/Tyr204) Antibody #9101#9101 s, Cell Signaling Technology), anti-$\beta_H$-spectrin (gift from G. Thomas, Pennsylvania State University, USA, 1:1000 dilution); anti-Par6 (gift from D. J. Montell, University of California Santa Barbara, USA, 1:500 dilution); anti-Mesh and anti-Tsp2A (gift from Mikio Furuse, National Institute for Physiological Sciences, Okazaki, Japan, 1:1000 dilution); anti-Pdm1 (gift from F. J. Diaz-Benjumea, Centre for Molecular Biology 'Severo Ochoa' (CBMSO), Spain, 1:1000 dilution); anti-Canoe (gift from M. Peifer, University of North Carolina, USA, 1:1000 dilution); anti-Sox21a (*Meng and Biteau, 2015*) (gift from B. Biteau, University of Rochester, USA, 1:1000 dilution); anti-Rab11 (gift from Akira Nakamura, Kumamoto University, Japan).

Other antibodies used: Chicken anti-GFP (Abcam, Cat. #ab13970, 1:1,000 dilution); Guinea pig anti-Myo7a (*Glowinski et al., 2014*)(gift from D. Godt, University of Toronto, Canada, 1:1000 dilution), Guinea pig anti-Shot (*Nashchekin et al., 2016*).

### Secondary antibodies

Alexa Fluor secondary antibodies (Invitrogen) were used at a dilution of 1:1,000.

Alexa Fluor 488 goat anti-mouse (#A11029), Alexa Fluor 488 goat anti-rabbit (#A11034), Alexa Fluor 488 goat anti-guinea pig (#A11073), Alexa Fluor 488 goat anti-chicken IgY (#A11039), Alexa Fluor 555 goat anti-mouse (#A21422), Alexa Fluor 555 goat anti-rabbit (#A21428), Alexa Fluor 568 goat anti-guinea pig (#A11075), Alexa Fluor 647 goat anti-mouse (#A21236), Alexa Fluor 647 goat anti-rabbit (#A21245). Only cross-adsorbed secondary antibodies were used in this study to eliminate the risk of cross-reactivity.

F-Actin was stained with phalloidin conjugated to Rhodamine (Invitrogen, Cat. #R415, 1:500 dilution).

## Immunofluorescence imaging

Images were collected on an Olympus IX81 (40×1.35 NA Oil UPlanSApo, 60×1.35 NA Oil UPlanSApo) using the Olympus FluoView software Version 3.1 and processed with Fiji (ImageJ).

## Sample processing for electron microscopy and TEM imaging

Samples were fixed in fixative (2% glutaraldehyde/2% formaldehyde in 0.05 M sodium cacodylate buffer pH7.4 containing 2 mM calcium chloride) overnight at 4 °C. After washing 5 x with 0.05 M sodium cacodylate buffer pH7.4, samples were osmicated (1% osmium tetroxide, 1.5% potassium

ferricyanide, 0.05 M sodium cacodylate buffer pH7.4) for 3 days at 4 °C. After washing 5 x in DIW (deionised water), samples were treated with 0.1% (w/v) thiocarbohydrazide/DIW for 20 minutes at room temperature in the dark. After washing 5 x in DIW, samples were osmicated a second time for 1 hour at RT (2% osmium tetroxide/DIW). After washing 5 x in DIW, samples were blockstained with uranyl acetate (2% uranyl acetate in 0.05 M maleate buffer pH5.5) for 3 days at 4 °C. Samples were washed 5 x in DIW and then dehydrated in a graded series of ethanol (50%/70%/95%/100%/100% dry) 100% dry acetone and 100% dry acetonitrile, 3 x in each for at least 5 min. Samples were infiltrated with a 50/50 mixture of 100% dry acetonitrile/Quetol resin (without BDMA) overnight, followed by 3 days in 100% Quetol (without BDMA). Then, the sample was infiltrated for 5 days in 100% Quetol resin with BDMA, exchanging the resin each day. The Quetol resin mixture is: 12 g Quetol 651, 15.7 g NSA, 5.7 g MNA and 0.5 g BDMA (all from TAAB). Samples were placed in embedding moulds and cured at 60 °C for 3 days.

Ultrathin sections (~70 nm) were cut on a Leica Ultracut E ultramicrotome and placed on bare 300 mesh copper TEM grids. Samples were viewed in a Tecnai G20 electron microscope (FEI/ThermoFisher SCientific) operated at 200 keV using a 20 µm objective aperture to enhance contrast. Images were acquired using an ORCA HR high resolution CCD camera (Advanced Microscopy Techniques Corp.).

### Generation of *FRT82B canoe[jc]* and *FRTG13 coracle[jc]* flies

We used the CRISPR/Cas9 method (*Bassett and Liu, 2014*) to generate null alleles of *canoe* and *coracle*. sgRNA was in vitro transcribed from a DNA template created by PCR from two partially complementary primers:

> forward primer:
> For *canoe*:5'-GAAATTAATACGACTCACTATA<u>GGGGCAAGGTAATGACGGAG</u>GTTTTAGAGCTAGAAATAGC-3';
> For coracle: 5'-GAAATTAATACGACTCACTATA<u>GAAGCTGGCCATGTACGGCG</u>GTTTTAGAGCTAGAAATAGC-3';
> reverse primer: 5'- AAAAGCACCGACTCGGTGCCACTTTTTCAAGTTGATAACGGACTAGCCTTATTTTAACTTGCTATTTCTAGCTCTAAAAC-3'.

The sgRNA was injected into *Act5c-Cas9; FRT82B* embryos to generate *canoe* null alleles, and into *Act5c-Cas9* embryos to generate *coracle* null alleles (*Port et al., 2014*). Putative *canoe* and *coracle* mutants in the progeny of the injected embryos were recovered, balanced, and sequenced. The *canoe[jc]* allele contains a small deletion around the CRISPR site, resulting in one missense mutation and a frameshift that leads to stop codon at amino acid 908 in the middle of the Dilute (DIL) domain, which is shared by all isoforms. No Canoe protein was detectable by antibody staining in both midgut and follicle cell clones. The *coracle[jc]* allele contains a 2 bp deletion around the CRISPR site, resulting in a frameshift that leads to stop codon at amino acid 225 in the middle of the FERM domain, which is shared by all isoforms. No Coracle protein was detectable by antibody staining (DSHB C615.16) in both midgut and follicle cell clones. The *coracle[jc]* allele was recombined with *FRT G13* to generate the *FRTG13 coracle[jc]* flies.

### Generation of the *mesh[R2]* flies

*FRT82B canoe[R2]* flies were crossed with empty *FRT82* flies for homologous recombination. The progenies were balanced then crossed with *FRT82B canoe[R10]* or *FRT82B mesh[f04955]* flies to identify the mesh null allele. Two homozygous lethal lines were obtained afterwards, named as *mesh[R2]*, both mutant cells gave the same phenotype as in adult midgut as *mesh[f04955]*, neither has any embryonic defect as seen in *canoe[R2]* nor lose the Canoe antibody staining in mutant follicle cells.

### Measuring ECad/Arm intensities in integrating cells at different integration stages

Fiji was used to measure membrane ECad, cortical Canoe, Coracle, Arm and UtrABD-GFP intensities in integrating enteroblasts at different stages as shown in *Figure 4*. We used Fiji plot line tool to mark the cell contours excluding the basal region and measured the fluorescence intensities of different proteins along the plot line. The x-axis corresponds to the position along the cell contour expressed as a distance in microns. Graphs were made using GraphPad Prism 9.

## Measuring cortical Canoe intensities in *mesh*[R2] mutant and wild-type cells

Fiji was used to measure cortical Canoe intensities in MARCM clones of *mesh*[R2] and neighbouring wild type integrating enteroblasts. We used the plot line tool to mark the cell contours excluding the basal region, so that the centre value of the plot line represents the apical side and both peripheral values represent the basal-lateral side. Canoe intensities along the plot line were divided by the maximum Canoe intensity in the sample to give the percentage value. The x-axis equals to the position along the cell contour expressed as a percentage of the full-length value.

## Measuring Sox21a nuclear intensity in integrating enteroblasts in different stages

*sqh::UtroABD-GFP* flies were heat shocked a day before dissection and heat-fixation. Rabbit anti-Sox21a was used to detect Sox21a nuclear signal. Sox21a nuclear intensity was measured in different stages of enteroblast integration relative to the apical actin signal using Fiji. The nuclear Sox21a level in enterocytes was measured in the same sample for comparison. Superplot was made using GraphPad Prism 9 software (*Lord et al., 2020*).

## Counting the number of cells of various septate junction mutants in different stages of integration processes

SJ mutant cells (MARCM or negatively marked) were classified into five stages: unpolarised cells, polarised cells, cells with PAC, cells with internal PAC and cells with apical domain based on the stained actin signal as diffused, actin patch apically, PAC shape actin, internal apicosome like, and apical brush border as apical domain. The number mutant cells at each stage was counted using Fiji and graph was made by GraphPad Prism 9.

## Acknowledgements

We thank Gurdon Institute Imaging Facility for microscopy and image analysis support, Karin Mueller and the Cambridge Advanced Imaging Centre for help with the transmission electron microscopy, Dr Jemima Burden (MRC_LMCB, UCL, London UK) for help with analysing TEM results, and members of the St Johnston laboratory for their advice and support. We are very grateful to Bruce Edgar, Tony Ip, Mark Peifer, Thomas Lecuit, Stefan Luschnig, Benjamin Klapholz, Nicholas Brown, Golnar Kolahgar, Mikio Furuse, Graham Thomas, Denise Montell, Fernando Díaz-Benjumea, Benoit Biteau, Akira Nakamura, Dorothea Godt, the Bloomington *Drosophila* stock centre, the Kyoto Stock Center and the Developmental Studies Hybridoma Bank for providing fly stocks and antibodies.

This work was supported by a Wellcome Principal Fellowship (207496) to D St J and core funding from the Wellcome Trust (203144) and Cancer Research UK (A24843).

## Additional information

### Funding

| Funder | Grant reference number | Author |
| --- | --- | --- |
| Wellcome Trust | Wellcome Principal Fellowship | Jia Chen<br>Daniel St Johnston |
| Cancer Research UK | CRUK Core funding | Jia Chen<br>Daniel St Johnston |
| Cancer Research UK | A24843 | Jia Chen<br>Daniel St Johnston |
| Wellcome Trust | 207496 | Jia Chen<br>Daniel St Johnston |
| Wellcome Trust | Wellcome Core funding | Jia Chen<br>Daniel St Johnston |

| Funder | Grant reference number | Author |
|---|---|---|
| Wellcome Trust | 203144 | Jia Chen<br>Daniel St Johnston |

The funders had no role in study design, data collection and interpretation, or the decision to submit the work for publication. For the purpose of Open Access, the authors have applied a CC BY public copyright license to any Author Accepted Manuscript version arising from this submission.

## Author contributions

Jia Chen, Conceptualization, Data curation, Formal analysis, Validation, Investigation, Visualization, Methodology, Writing - original draft; Daniel St Johnston, Conceptualization, Resources, Supervision, Funding acquisition, Visualization, Project administration, Writing - review and editing

## Author ORCIDs

Jia Chen http://orcid.org/0000-0003-0080-0748
Daniel St Johnston http://orcid.org/0000-0001-5582-3301

## Decision letter and Author response

Decision letter https://doi.org/10.7554/eLife.76366.sa1
Author response https://doi.org/10.7554/eLife.76366.sa2

# Additional files

## Supplementary files

• Transparent reporting form

## Data availability

All data analysed during this study are included in the manuscript and supporting file. Source Data files have been provided for Figures 1, 2, 5 and 8.

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
