## [Editor Report]

This paper addresses a fundamental cell biological question and will be of interest to a wide range of researchers including those in the fields of cell, development, stem cell and cancer biology. The finding that newborn cells (enterocytes) in the *Drosophila* midgut form a pre-apical compartment containing a fully-formed brush border prior to integrating into the epithelium, which is described in great detail, is highly novel and exciting. This work highlights similarities between fly midgut polarization and lumen formation in mammalian epithelial cysts.

---

## [Decision Letter]

**Decision letter after peer review:**

Thank you for submitting your article "de novo apical domain formation inside the *Drosophila* adult midgut epithelium" for consideration by *eLife*. Your article has been reviewed by 3 peer reviewers, one of whom is a member of our Board of Reviewing Editors, and the evaluation has been overseen by Utpal Banerjee as the Senior Editor. The following individual involved in review of your submission has agreed to reveal their identity: Bruce A Edgar (Reviewer #3).

Essential revisions:

1. The final step in the process, "fusion" of the pre-apical compartment (PAC) with the gut lumen is still somewhat mysterious, and there is little comment about it or data showing the final step. For instance, what happens to the membranes apical to the PAC? Do they dissolve or retract? If they retract, which cells get them? What is happening in Figure 8A,C,E, S7D? Is that a detachment phenotype or an integration phenotype? Are the majority of cells unpolarized due to loss of integrin attachment rather than failure to form an AMIS/PAC?

Is the model in Figure 6 supported by EM data – can you see a region where there is brush border and separation of cells? Supplementing Figure 3 with corresponding EM images would greatly aid the reader in interpreting the data and strengthen the model.

2. Role for the septate junction proteins/septate junction: The role of the septate junction proteins Mesh and Coracle, which are used interchangeably, should be clarified. Do they colocalize? Is their localization interdependent (as demonstrated for Mesh, Tsp2a and Ssk in Figure 7)? What is the phenotype of enteroblasts mutant for cora? If Cora is to be used as a readout for the localization of septate junction components, then staining for Cora/Mesh and/or Cora/SSk or Tsp2a should be shown. Does the localization of septate junction protein really correlate with the presence of septate junctions? What is the status of the septate junction itself, e.g. in the cells shown in Figure 2F?

3. The description of the phenotype resulting from lack of individual septate junction proteins stays somewhat vague. When exactly do things go wrong? It is hard to know what to conclude from this data about the role of the septate junction components in PAC formation. To help the reader evaluate this data, the authors should include information on the age of these MARCM clones and the numbers of cells per clone.

*Reviewer #1 (Recommendations for the authors):*

1. The data show the distribution and the role of septate junction proteins in this process. This led the authors to conclude that septate junctions play an important role. But does the localisation of septate junction protein really correlate with the presence of septate junctions? What is the status of the septate junction itself, e.g. in the cells shown in Figure 2F? And in an earlier stage?

2. In most of the cartoons it is not indicated whether a membrane belongs to the enteroblast or to an enterocyte, and in most cases this is not a problem. However, to better understand the processes going on at the apical pole, it would have been helpful to know which membrane belongs to which cell.

3. The description of the phenotype resulting from lack of individual septate junction proteins stays somewhat vague. When exactly do things go wrong? The authors conclude (page 25) that "… lack of septate junctions are not absolutely required for apical-basal polarity in integrating enteroblasts, ….". Can the authors exclude perdurance of the proteins in some cells?

4. I find it confusing that the authors used the term "AMIS", which, as they write, is clearly defined by Bryant et al., 2010, in MDCK cells and includes a midbody. This is clearly not the case in the system used here. Why not use a different term, e.g. incipient apical membrane?

5. The authors talk about the "AMIS" and the "PAC". When exactly does the AMIS turn into the PAC? And if the PAC is a membrane compartment: why can it be closed or open (Figure 2I)? In my understanding, the space/lumen above the PAC can be closed or open.

6. Discussion, page 27: The text reads: "… indicating that enteroblasts go through a mesenchymal stage." This suggests that they have been epithelial cells before, which I think is not the case (though they are polarised). In contrast, as pointed out, the embryonic midgut precursors undergo a clear EMT.

7. Discussion, page 28: It reads: "In mammalian epithelial cysts, the site of AMIS is defined by the position of the midbody…" I think this is only true for MDCK cells, and may be different in other systems, e.g. in cysts derived from stem cells.

8. Last sentence of the discussion: In fact, there is a similar system, the *Drosophila* Malpighian tubules, where individual cells, the Stellate cells, integrate from the basal side into an existing epithelium (Campbell et al. 2010). A comparison with this system would be nice to mention.

*Reviewer #3 (Recommendations for the authors):*

Overall, I thought this paper was exceptionally well done. It is packed with novel observations and the many conclusions are well supported by the data. A large number of markers are tracked and some are functionally tested. It was also easy to follow and the summary diagrams are clear and helpful. The novelty and significance are high. It does not require much revision. I have these comments for the authors:

1. The analysis is based on the ordering of different samples in developmental time by morphology and differentiation markers. Although this is a standard approach, and technically necessary, the authors should note cases where their analysis was ambiguous. Sufficient sample numbers are provided to support many conclusions about the order of events, but nevertheless inferences about progression needed to be made. The assumptions these inferences were based on need to be clearly stated. Please review the manuscript for this issue and amend the text as needed.

2. Although the order of steps is inferred, the authors do not comment on how long the process of EB/EC integration into the epithelium actually takes, and under what conditions (for instance with or without heat shock). Comments on the timing of events would add to the value of the manuscript.

3. The final step in the process, "fusion" of the pre-apical compartment (PAC) with the gut lumen is still somewhat mysterious, and there is little comment about it or data showing the final step. For instance, what happens to the membranes apical to the PAC? Do they dissolve or retract? If they retract, which cells get them? Please comment on this.

4. Although I don't doubt the general conclusions, the smooth septate junction mutant phenotypes are quite messy. To help the reader evaluate this data, the authors should include information on the age of these MARCM clones and the numbers of cells per clone.

5. A parallel study with similar conclusions but fewer descriptions of protein localizations has appeared on BioRxiv and is likely to be published soon (https://doi.org/10.1101/2021.09.19.457819). So as to not confuse the field, the authors should try to use similar terms and acronyms in their two studies, and discussion of discrepant points and cross citations would be helpful.

---

## [Author Response]

Essential revisions:1. The final step in the process, "fusion" of the pre-apical compartment (PAC) with the gut lumen is still somewhat mysterious, and there is little comment about it or data showing the final step. For instance, what happens to the membranes apical to the PAC? Do they dissolve or retract? If they retract, which cells get them?

We propose that the EC-EC septate junctions above the internal lumen are disassembled. This allows the membranes of the two ECs to separate, so that the internal lumen above the PAC becomes continuous with the gut lumen and the PAC becomes the apical domain of the newly-integrated enterocyte. Most EBs integrate beneath tricellular junctions and this TCJ must also be disassembled to allow the new enterocyte to emerge. Thus, we propose that the septate and tricellular junctions above the pre-enterocyte dissolve and the enterocyte membranes retract.

We have now added the following text to the results:

"Finally, the septate junctions above the internal lumen disassemble, allowing the membranes of the overlying enterocytes to separate, so that the lumen becomes continuous with the gut lumen. At this point the PAC becomes the apical domain of the newly-integrated enterocyte (Figure 6). The mechanisms that drive the disassembly of the overlying septate junctions are not known, but it is worth bearing in mind that most EBs integrate beneath a tricellular junction. This tricellular junction must therefore also be disassembled to allow the new enterocyte to emerge on the apical surface, and three new tricellular junctions must eventually form where the new enterocyte meets two of the adjacent enterocytes. Indeed, the disassembly of the overlying tricellular junction may be the first event that triggers the emergence of the new enterocyte."

We have also added a new top view model as Figure 6B, showing the proposed final step of the fusion of the separate lumen with the gut lumen, dissolved tri-cellular junction (TCJ) and SJs in the neighbouring ECs and the newly formed TCJs and SJs between the integrated EC and neighbouring ECs.

What is happening in Figure 8A,C,E, S7D? Is that a detachment phenotype or an integration phenotype? Are the majority of cells unpolarized due to loss of integrin attachment rather than failure to form an AMIS/PAC?

Cells mutant for septate junction proteins do not detach from the basement membrane and still localise Talin basally, as illustrated by the new panel we have added (Figure8—figure supplement 1N), showing Talin localisation in *Tsp2a* mutant cell.

We have added the following sentences to the Results, explaining these points:

"Because the mutant cells remain trapped beneath enterocyte-enterocyte septate junctions, they accumulate in the basal region of the epithelium, with new EBs derived from the same mutant ISC forming beneath them and reducing their contact with the basement membrane (Figure 8A)."

"The majority of cells mutant for septate junction components fail to polarise or form an AMIS, although they form normal lateral and basal domains, as shown by the normal localisation of the basal integrin signalling component, Talin (Figure 8—figure supplement 1N)."

Is the model in Figure 6 supported by EM data – can you see a region where there is brush border and separation of cells? Supplementing Figure 3 with corresponding EM images would greatly aid the reader in interpreting the data and strengthen the model.

We think that AJ clearing and membrane separation is a brief process that is quickly followed by the separation of the apical and junctional proteins and apical secretion at the AMIS to form the PAC. We have not captured this stage in our EM images, but have many other examples that show this step (e.g Figure 4C and Figure 8F). Another example is shown in Author response image 1.

**Author response image 1. sa2fig1:** 

2. Role for the septate junction proteins/septate junction: The role of the septate junction proteins Mesh and Coracle, which are used interchangeably, should be clarified. Do they colocalize? Is their localization interdependent (as demonstrated for Mesh, Tsp2a and Ssk in Figure 7)? What is the phenotype of enteroblasts mutant for cora? If Cora is to be used as a readout for the localization of septate junction components, then staining for Cora/Mesh and/or Cora/SSk or Tsp2a should be shown.

The reason we used mainly Coracle as a marker for the septate junctions is that Mesh and Tsp2A localise to the basal labyrinth as well as to the septate junctions which is not aesthetically pleasing. We have now added new panels to Figure 3—figure supplement 1E&F showing the colocalization of Cora with Mesh/Tsp2a at the septate junctions and during the crucial stage of PAC formation.

Additional Results:

"Coracle is a peripheral septate junction protein whose localisation depends on the structural septate junction components such as Mesh/Ssk/Tsp2a (Chen et al., 2018; Izumi et al., 2016, 2012). Cora antibody staining provides a clearer marker for the septate junctions than Mesh or Tsp2a antibody staining, because the latter also label the basal labyrinth (Figure 3—figure supplement 1E&F). To determine whether Cora is required for PAC formation or epithelial polarity in the adult midgut, we generated a null mutant allele with a premature stop codon in FERM domain using CRISPR. Cells mutant for this allele, *cora*^jc^, or a second *cora* null allele, *cora*^5^, can form a PAC, septate junctions and a full apical domain, indicating that Cora is also not required for enteroblast integration or enterocyte polarity (Figure 7F&G, Figure 7—figure supplement 1E-H).

Additional Materials and methods:

We used the CRISPR/Cas9 method (Bassett and Liu, 2014) to generate null alleles of *canoe* and *coracle*. sgRNA was in vitro transcribed from a DNA template created by PCR from two partially complementary primers:

forward primer:

For coracle:

5′-GAAATTAATACGACTCACTATAGAAGCTGGCCATGTACGGCGGTTTTAGAGCTAGAAATAGC-3′;

The sgRNA was injected into…*Act5c-Cas9* embryos to generate *coracle* null alleles (Port et al., 2014). Putative…*coracle* mutants in the progeny of the injected embryos were recovered, balanced, and sequenced. …The *coracle*^jc^ allele contains a 2bp deletion around the CRISPR site, resulting in a frameshift that leads to stop codon at amino acid 225 in the middle of the FERM domain, which is shared by all isoforms. No Coracle protein was detectable by antibody (DSHB C615.16) staining in both midgut and follicle cell clones. The *coracle*^jc^ allele was recombined with *FRT G13* to make the *FRTG13 coracle*^jc^ flies.

Does the localization of septate junction protein really correlate with the presence of septate junctions? What is the status of the septate junction itself, e.g. in the cells shown in Figure 2F?

The correlation between the presence of septate junction proteins and the morphological formation of the septate junction itself has been demonstrated in several papers (Izumi et al., 2021, 2016, 2012). We could not resolve the septate junctions in our EM on wild type enterocytes, and are therefore unable to answer this question.

3. The description of the phenotype resulting from lack of individual septate junction proteins stays somewhat vague. When exactly do things go wrong? It is hard to know what to conclude from this data about the role of the septate junction components in PAC formation.

We have re-written the conclusion to this section to make it clearer and now explicitly mention perdurance:

"The majority of cells mutant for septate junction components fail to polarise or form an AMIS, although they form normal lateral and basal domains, as shown by the normal localisation of the basal integrin signalling component, Talin (Figure 8—figure supplement 1N). This contrasts with the initial apical-basal polarisation of ISCs and quiescent enteroblasts, which are not affected in *mesh*, *Tsp2a* and *ssk* mutant clones. The mutant clones also have similar sizes to the control empty FRT82B clones, indicating that they have no effect on stem cell behaviour or division rate (Figure 8—figure supplement 1O). Thus, the septate junction proteins are not involved in the initial polarisation of ISCs and quiescent enteroblasts, but they are required for the apical-basal polarisation that gives rise to the AMIS in most cells. This is distinct from the role of integrin adhesion complex, which is required for apical-basal polarity in ISCs, quiescent enteroblasts and activated enteroblasts (Figure 8—figure supplement 2D-F).

Some mutant enteroblasts still polarise to form an AMIS-like structure in the absence of septate junction proteins, since 5-20% of mutant cells still localise actin, Par-6 and Rab11 apically (Arrows in Figure 8E-H), lose their apical Adherens junctions and detach their apical membrane from the overlying wild-type enterocytes (Figure 8D&F). In the case of *mesh* and *Tsp2A,* this AMIS-like structure never gives rise to a PAC. The low frequency of AMIS formation in these mutants may be due to partial rescue of the phenotype by the perdurance of the wild-type proteins after clone induction. Alternatively, the formation of the AMIS may not absolutely require the septate junction proteins, but only occur very inefficiently in their absence."

To help the reader evaluate this data, the authors should include information on the age of these MARCM clones and the numbers of cells per clone.

We have added these data as Figure 8—figure supplement 1O:

We have also added the following sentence to the main text:

“The mutant clones also have similar sizes to the control empty FRT82B clones, indicating that they have no effect on stem cell behaviour or division rate (Figure 8—figure supplement 1O).”

As suggested, we have also included a sentence describing the age of the clones to the material and methods:

“To generate homozygous mutant clones of *mesh*, *ssk* and *Tsp2a*, flies were heat shocked as larvae (from L2 until eclosion) and dissected 4 days after eclosion.

Reviewer #1 (Recommendations for the authors):1. The data show the distribution and the role of septate junction proteins in this process. This led the authors to conclude that septate junctions play an important role. But does the localisation of septate junction protein really correlate with the presence of septate junctions? What is the status of the septate junction itself, e.g. in the cells shown in Figure 2F? And in an earlier stage?

The correlation between the presence of septate junction proteins and the morphological formation of the septate junction itself has been demonstrated in several papers (Izumi et al., 2021, 2016, 2012). , We could not resolve the septate junctions in our EM on wild type enterocytes, and are therefore unable to answer this question.

2. In most of the cartoons it is not indicated whether a membrane belongs to the enteroblast or to an enterocyte, and in most cases this is not a problem. However, to better understand the processes going on at the apical pole, it would have been helpful to know which membrane belongs to which cell.

We have changed the cartoon in Figure 2I, by separating the membrane layers between the pre-EC and neighbouring ECs.

We have also changed the figure legend as follows: (I) A model for enteroblast integration in which a “closed” lumen above the PAC precedes an “open” lumen. The “closed” lumen stage represents the pre-EC with a PAC forming underneath the septate junction between the neighbouring enterocytes (purple), creating an isolated, closed lumen inside the epithelial layer. The cap over the “closed” lumen comes from the neighbouring enterocytes. Adherens junctions form between pre-EC (light blue) and neighbouring enterocytes. New septate junctions also form between the pre-EC and the adjacent enterocytes (purple). The “open” lumen represents a fully-developed enterocyte after the lumen has fused with the gut lumen, turning the PAC into the apical domain. To simplify the cartoons in the following figures, we combine the membranes between pre-EC and neighbouring ECs into one line.

3. The description of the phenotype resulting from lack of individual septate junction proteins stays somewhat vague. When exactly do things go wrong? The authors conclude (page 25) that "… lack of septate junctions are not absolutely required for apical-basal polarity in integrating enteroblasts, ….". Can the authors exclude perdurance of the proteins in some cells?

We have re-written the conclusion to this section to make it clearer and now explicitly mention perdurance:

"The majority of cells mutant for septate junction components fail to polarise or form an AMIS, although they form normal lateral and basal domains, as shown by the normal localisation of the basal integrin signalling component, Talin (Figure 8—figure supplement 1N). This contrasts with the initial apical-basal polarisation of ISCs and quiescent enteroblasts, which are not affected in *mesh*, *Tsp2a* and *ssk* mutant clones. The mutant clones also have similar sizes to the control empty FRT82B clones, indicating that they have no effect on stem cell behaviour or division rate (Figure 8—figure supplement 1O). Thus, the septate junction proteins are not involved in the initial polarisation of ISCs and quiescent enteroblasts, but they are required for the apical-basal polarisation that gives rise to the AMIS in most cells. This is distinct from the role of integrin adhesion complex, which is required for apical-basal polarity in ISCs, quiescent enteroblasts and activated enteroblasts (Figure 8—figure supplement 2D-F).

Some mutant enteroblasts still polarise to form an AMIS-like structure in the absence of septate junction proteins, since 5-20% of mutant cells still localise actin, Par-6 and Rab11 apically (Arrows in Figure 8E-H), lose their apical Adherens junctions from this region and detach their apical membrane from the overlying wild-type enterocytes (Figure 8D&F). In the case of *mesh* and *Tsp2A,* this AMIS-like structure never gives rise to a PAC. The low frequency of AMIS formation in these mutants may be due to partial rescue of the phenotype by the perdurance of the wild-type proteins after clone induction. Alternatively, the formation of the AMIS may not absolutely require the septate junction proteins, but only occur very inefficiently in their absence."

4. I find it confusing that the authors used the term "AMIS", which, as they write, is clearly defined by Bryant et al., 2010, in MDCK cells and includes a midbody. This is clearly not the case in the system used here. Why not use a different term, e.g. incipient apical membrane?

As pointed out in this reviewer's question 7, the midbody may position the AMIS in some contexts, e.g. MDCK cysts in 3D culture, but does not define the AMIS (Apical Membrane Initiation Site). In other systems, such as the zebrafish neural tube, the midbody is not required for apical domain formation, since it still forms when cell division is blocked (Symonds et al., 2020). Thus, we want to keep the term “AMIS” in our paper to highlight the similarity between apical domain formation in the fly midgut epithelium and vertebrate epithelia.

5. The authors talk about the "AMIS" and the "PAC". When exactly does the AMIS turn into the PAC? And if the PAC is a membrane compartment: why can it be closed or open (Figure 2I)? In my understanding, the space/lumen above the PAC can be closed or open.

We propose that the AMIS turns into a PAC when the new septate junctions form laterally and the apical membrane starts to develop. This is preceded by clearing of the AJs from the apical membrane, separation of the pre-EC plasma membrane from the membranes of the overlying enterocytes and fluid secretion to form a lumen.

We agree with the referee that it is the lumen that is open or closed and have changed the text accordingly.

6. Discussion, page 27: The text reads: "… indicating that enteroblasts go through a mesenchymal stage." This suggests that they have been epithelial cells before, which I think is not the case (though they are polarised). In contrast, as pointed out, the embryonic midgut precursors undergo a clear EMT.

We do not think that the term mesenchymal implies that the cells were previously epithelial, but agree with the referee that the parallel with the embryonic midgut precursors is overstated. We have therefore re-written this section to make clear that we are only referring to the reversible loss of polarity (not specifically epithelial polarity):

"This reveals that growing enteroblasts go through a phase where they lack a distinct apical domain, in contrast to ISCs and quiescent enteroblasts, which are polarised and localise Canoe and Par-6 apically. These unpolarised enteroblasts are likely to correspond to the protrusive, migratory state of activated enteroblasts reported by Antonella et al., indicating that growing enteroblasts go through a mesenchymal stage (Antonello et al., 2015). Enteroblasts then re-polarise once they reach the septate junctions between the overlying enterocytes. Thus, the adult enteroblasts transiently lose polarity and become migratory before developing epithelial polarity, much like the embryonic midgut precursors, which undergo an epithelial to mesenchymal transition when they delaminate from the primary epithelium and become migratory, before re-polarising in contact with the visceral mesoderm to form the embryonic midgut epithelium (Campbell et al., 2011; Pitsidianaki et al., 2021; Tepass and Hartenstein, 1994)."

7. Discussion, page 28: It reads: "In mammalian epithelial cysts, the site of AMIS is defined by the position of the midbody…" I think this is only true for MDCK cells, and may be different in other systems, e.g. in cysts derived from stem cells.

We agree with the reviewer and have changed the text as follows:

"In most mammalian epithelial cysts, the site of the AMIS is defined by the position of the midbody formed during the last cell division, whereas enteroblasts are postmitotic and are derived from intestinal stem cell divisions that may have occurred several days earlier (Mangan et al., 2016; Rojas Villa et al., 2019; Tang et al., 2021).”

8. Last sentence of the discussion: In fact, there is a similar system, the *Drosophila* Malpighian tubules, where individual cells, the Stellate cells, integrate from the basal side into an existing epithelium (Campbell et al. 2010). A comparison with this system would be nice to mention.

We now mention the Stellate cells in the introduction, since they integrate in a similar way to the multi-ciliate cells in the *Xenopus* epidermis (Campbell et al., 2010).

Reviewer #3 (Recommendations for the authors):Overall, I thought this paper was exceptionally well done. It is packed with novel observations and the many conclusions are well supported by the data. A large number of markers are tracked and some are functionally tested. It was also easy to follow and the summary diagrams are clear and helpful. The novelty and significance are high. It does not require much revision. I have these comments for the authors:1. The analysis is based on the ordering of different samples in developmental time by morphology and differentiation markers. Although this is a standard approach, and technically necessary, the authors should note cases where their analysis was ambiguous. Sufficient sample numbers are provided to support many conclusions about the order of events, but nevertheless inferences about progression needed to be made. The assumptions these inferences were based on need to be clearly stated. Please review the manuscript for this issue and amend the text as needed.

We have added a sentence when describing the steps in the main-text as “Imaging heat shocked flies expressing *sqh::UtrABD-GFP* or *Fim-GFP* and stained for Canoe and Cora revealed three distinct stages of PAC formation. Based on cell size and marker protein localisation, we infer the follow sequence of steps.” We have also added one sentence to the discussion:

"This sequence of events has been inferred from ordering images of many fixed samples based on cell size and the localisation of various proteins, but it will be important to corroborate this sequence by imaging enteroblast integration in living midguts."

2. Although the order of steps is inferred, the authors do not comment on how long the process of EB/EC integration into the epithelium actually takes, and under what conditions (for instance with or without heat shock). Comments on the timing of events would add to the value of the manuscript.

We have performed a new experiment in which we over-express Sox21a using esg[ts]>GFP to induce enteroblast differentiation, and then count the number of GFP^+ve^ cells without a PAC, with a PAC and with full apical domain at different time points. This gives us a rough estimate of 1 day as the average time it takes for an activated EB to become a pre-enterocyte with a PAC and for a pre-enterocyte to reach the gut lumen and become an enterocyte.

We have summarised these results in Figure 5—figure supplement 1C.

We have also added the following to the main text:

"To estimate the time taken for enteroblasts to progress to pre-enterocytes with a PAC, and for pre-enterocytes become to enterocytes, we induced enterocyte differentiation by over-expressing *UAS-Sox21a* under the control of *esg[ts]-Gal4* and counted the number of GFP^+ve^ cells without a PAC or apical domain, with a PAC and with a full apical domain at different time points after induction (Chen et al., 2016; Meng and Biteau, 2015; Zhai et al., 2017). 17 hours after shifting the flies to 25ºC to inactivate Gal80^ts^, almost no GFP^+ve^ cells had progressed to pre-EC with a PAC (0.1%) or EC (1%), and these few cells probably started to differentiate before Sox 21a induction. 24 hours later, 10% of the GFP^+ve^ cells had developed into pre-ECs with a PAC and 20% had become ECs (Figure 5—figure supplement 1B&C). After an additional 24 hours, the number of cells with a PAC fell to 1%, whereas 50% were ECs. Assuming that it takes 12-17 hours to induce high levels of Sox21a expression, these results suggest that most activated EBs take about 24 hours to develop into a pre-EC with a PAC and a further 24 hours to differentiate into a mature EC, although some cells differentiate faster. This time frame is in agreement with a previous study using similar approaches to accelerate differentiation (Rojas Villa et al., 2019)and a recent live imaging study tracing the enteroblast to enterocyte transition (Tang et al., 2021). This experiment also indicates that down-regulation of Sox21a is not essential for enteroblast to pre-enterocyte differentiation, since enteroblasts overexpressing Sox21a still from a PAC (Figure 5—figure supplement 1B).”

3. The final step in the process, "fusion" of the pre-apical compartment (PAC) with the gut lumen is still somewhat mysterious, and there is little comment about it or data showing the final step. For instance, what happens to the membranes apical to the PAC? Do they dissolve or retract? If they retract, which cells get them? Please comment on this.

We propose that the EC-EC septate junctions above the internal lumen are disassembled. This allows the membranes of the two ECs to separate, so that the internal lumen above the PAC becomes continuous with the gut lumen and the PAC becomes the apical domain of the newly-integrated enterocyte. Most EBs integrate beneath tricellular junctions and this TCJ must also be disassembled to allow the new enterocyte to emerge. Thus, we propose that the septate and tricellular junctions above the pre-enterocyte dissolve and the enterocyte membranes retract.

We have now added the following text to the results:

“Finally, the septate junctions above the internal lumen disassemble, allowing the membranes of the overlying enterocytes to separate, so that the lumen becomes continuous with the gut lumen. At this point the PAC becomes the apical domain of the newly-integrated enterocyte (Figure 6). The mechanisms that drive the disassembly of the overlying septate junctions are not known, but it is worth bearing in mind that most EBs integrate beneath a tricellular junction. This tricellular junction must therefore also be disassembled to allow the new enterocyte to emerge on the apical surface, and three new tricellular junctions must eventually form where the new enterocyte meets two of the adjacent enterocytes. Indeed, the disassembly of the overlying tricellular junction may be the first event that triggers the emergence of the new enterocyte.”

We have also added a new top view model as Figure 6B, showing the proposed final step of the fusion of the separate lumen with the gut lumen, dissolved tri-cellular junction (TCJ) and SJs in the neighbouring ECs and the newly formed TCJs and SJs between the integrated EC and neighbouring ECs.

4. Although I don't doubt the general conclusions, the smooth septate junction mutant phenotypes are quite messy. To help the reader evaluate this data, the authors should include information on the age of these MARCM clones and the numbers of cells per clone.

We have now completely re-written this section and have added the data on the number of cells in the clones as Figure 8—figure supplement 1O:

As suggested, we have also included a sentence describing the age of the clones to the material and methods:

“To generate homozygous mutant clones of *mesh*, *ssk* and *Tsp2a*, flies were heat shocked as larvae (from L2 until eclosion) and dissected 4 days after eclosion.”

5. A parallel study with similar conclusions but fewer descriptions of protein localizations has appeared on BioRxiv and is likely to be published soon (https://doi.org/10.1101/2021.09.19.457819). So as to not confuse the field, the authors should try to use similar terms and acronyms in their two studies, and discussion of discrepant points and cross citations would be helpful.

We thank the reviewer for pointing this out. We have talked to the corresponding author of this BioRxiv manuscript and have agreed to use the same terms and acronyms (e.g pre-assembled apical compartment; PAC). We have cited their paper in our revised version as follows:

"Our results are entirely consistent with another recent study of enteroblast integration in the adult midgut, which reached very similar conclusions (Moreno-Roman et al., 2021).”

References

Antonello ZA, Reiff T, Ballesta-Illan E, Dominguez M. 2015. Robust intestinal homeostasis relies on cellular plasticity in enteroblasts mediated by miR-8-Escargot switch. *Embo j* 34:2025–2041. doi:10.15252/embj.201591517

Bassett A, Liu JL. 2014. CRISPR/Cas9 mediated genome engineering in *Drosophila*. *Methods* 69:128–136. doi:10.1016/j.ymeth.2014.02.019

Campbell K, Casanova J, Skaer H. 2010. Mesenchymal-to-epithelial transition of intercalating cells in *Drosophila* renal tubules depends on polarity cues from epithelial neighbours. *Mech Dev* 127:345–57. doi:10.1016/j.mod.2010.04.002

Campbell K, Whissell G, Franch-Marro X, Batlle E, Casanova J. 2011. Specific GATA Factors Act as Conserved Inducers of an Endodermal-EMT. *Developmental Cell* 21:1051–1061. doi:10.1016/j.devcel.2011.10.005

Chen HJ, Li Q, Nirala NK, Ip YT. 2020. The Snakeskin-Mesh Complex of Smooth Septate Junction Restricts Yorkie to Regulate Intestinal Homeostasis in *Drosophila*. *Stem Cell Reports* 14:828–844. doi:10.1016/j.stemcr.2020.03.021

Chen J, Sayadian AC, Lowe N, Lovegrove HE, St Johnston D. 2018. An alternative mode of epithelial polarity in the *Drosophila* midgut. *PLoS Biology* 16. doi:10.1371/journal.pbio.3000041

Chen J, Xu N, Huang H, Cai T, Xi R. 2016. A feedback amplification loop between stem cells and their progeny promotes tissue regeneration and tumorigenesis. *ELife*. doi:10.7554/*eLife*.14330.001

Choi NH, Lucchetta E, Ohlstein B. 2011. Nonautonomous regulation of *Drosophila* midgut stem cell proliferation by the insulin-signaling pathway. *Proc Natl Acad Sci U S A* 108:18702–18707. doi:10.1073/pnas.1109348108

Hung R-J, Hu Y, Kirchner R, Liu Y, Xu C, Comjean A, Gopal Tattikota S, Li F, Song W, Ho Sui S, Perrimon N. 2020. A cell atlas of the adult *Drosophila* midgut. *Proc Natl Acad Sci U S A* 117:1514–1523. doi:10.1073/pnas.1916820117

Izumi Y, Furuse K, Furuse M. 2021. The novel membrane protein Hoka regulates septate junction organization and stem cell homeostasis in the *Drosophila* gut. *J Cell Sci* 134. doi:10.1242/jcs.257022

Izumi Y, Furuse K, Furuse M. 2019. Septate junctions regulate gut homeostasis through regulation of stem cell proliferation and enterocyte behavior in *Drosophila*. *J Cell Sci* 132. doi:10.1242/jcs.232108

Izumi Y, Motoishi M, Furuse K, Furuse M. 2016. A tetraspanin regulates septate junction formation in *Drosophila* midgut. *J Cell Sci* 129:1155–1164. doi:10.1242/jcs.180448

Izumi Y, Yanagihashi Y, Furuse M. 2012. A novel protein complex, Mesh-Ssk, is required for septate junction formation in the *Drosophila* midgut. *J Cell Sci* 125:4923–4933. doi:10.1242/jcs.112243

Liang J, Balachandra S, Ngo S, O’Brien LE. 2017. Feedback regulation of steady-state epithelial turnover and organ size. *Nature* 548:588–591. doi:10.1038/nature23678

Mangan AJ, Sietsema D v, Li D, Moore JK, Citi S, Prekeris R. 2016. Cingulin and actin mediate midbody-dependent apical lumen formation during polarization of epithelial cells. *Nat Commun* 7:12426. doi:10.1038/ncomms12426

Meng FW, Biteau B. 2015. A Sox Transcription Factor Is a Critical Regulator of Adult Stem Cell Proliferation in the *Drosophila* Intestine. *Cell Rep* 13:906–914. doi:10.1016/j.celrep.2015.09.061

Moreno-Roman P, Su Y-H, Galenza A, Acosta L, Debec A, Guichet A, Knapp J-M, Kizilyaprak C, Humbel BM, Kolotuev I, Lucy &, O’brien E. 2021. Progenitor cell integration into a barrier epithelium during adult organ turnover. *bioRxiv*. doi:10.1101/2021.09.19.457819

Pitsidianaki I, Morgan J, Adams J, Campbell K. 2021. Mesenchymal-to-epithelial transitions require tissue-specific interactions with distinct laminins. *Journal of Cell Biology* 220. doi:10.1083/jcb.202010154

Port F, Chen HM, Lee T, Bullock SL. 2014. Optimized CRISPR/Cas tools for efficient germline and somatic genome engineering in *Drosophila*. *Proc Natl Acad Sci U S A* 111:E2967-76. doi:10.1073/pnas.1405500111

Rojas Villa SE, Meng FW, Biteau B. 2019. zfh2 controls progenitor cell activation and differentiation in the adult *Drosophila* intestinal absorptive lineage 15:e1008553. doi:10.1371/journal.pgen.1008553

Symonds AC, Buckley CE, Williams CA, Clarke JDW. 2020. Coordinated assembly and release of adhesions builds apical junctional belts during de novo polarisation of an epithelial tube. *Development* 147. doi:10.1242/dev.191494

Tang R, Qin P, Liu X, Wu S, Yao R, Cai G, Gao J, Wu Y, Guo Z. 2021. Intravital imaging strategy FlyVAB reveals the dependence of *Drosophila* enteroblast differentiation on the local physiology. *Communications Biology* 4:1223. doi:10.1038/s42003-021-02757-z

Tepass U, Hartenstein V. 1994. Epithelium formation in the *Drosophila* midgut depends on the interaction of endoderm and mesoderm. *Development* 120:579–590. doi:10.1242/dev.120.3.579

Xu C, Tang HW, Hung RJ, Hu Y, Ni X, Housden BE, Perrimon N. 2019. The Septate Junction Protein Tsp2A Restricts Intestinal Stem Cell Activity via Endocytic Regulation of aPKC and Hippo Signaling. *Cell Rep* 26:670-688.e6. doi:10.1016/j.celrep.2018.12.079

Zhai Z, Boquete JP, Lemaitre B. 2017. A genetic framework controlling the differentiation of intestinal stem cells during regeneration in *Drosophila*. *PLoS Genet* 13:e1006854. doi:10.1371/journal.pgen.1006854